# Bidirectional Reverse Contrastive Distillation for Progressive Multi-Level Graph Anomaly Detection

## Abstract

Graph Anomaly Detection (GAD) faces the challenge of identifying irregular patterns across multiple structural scales while maintaining computational efficiency for real-world deployment. Existing knowledge distillation approaches rely on unidirectional teacher-student alignment, producing brittle embeddings that fail to establish robust decision boundaries between normal and anomalous patterns. We introduce RECODISTILL, a unified framework that combines bidirectional contrastive learning with progressive checkpoint-based distillation using a single teacher network. Our approach simultaneously optimizes two complementary objectives: (1) attracting student embeddings toward clean teacher representations while (2) repelling them from structured multi-scale noisy teacher outputs. We develop a dynamic curriculum mechanism that selects optimal teacher checkpoints based on complexity-compatibility trade-offs, progressing from local to global semantics. Unlike existing methods requiring multiple teacher networks, RECODISTILL achieves superior efficiency through single-teacher architecture while maintaining state-of-the-art performance. Evaluation on 14 benchmark datasets demonstrates that RECODISTILL achieves the best detection accuracy (88.93% AUROC on Amazon, 89.80% on BM-MN), superior zero-shot transfer performance across 9 out of 12 cross-task scenarios, and substantial computational efficiency improvements. Our theoretical analysis provides convergence guarantees and generalization properties, establishing RECODISTILL as the first computationally efficient GAD framework to unify bidirectional contrastive learning with progressive distillation.

## 1 Introduction

Graph Anomaly Detection (GAD) is a fundamental problem in machine learning with critical applications spanning financial fraud detection (Zhang et al., 2024; Gao et al., 2023), misinformation propagation analysis in social networks, and cybersecurity threat identification (Liu et al., 2022; Dou et al., 2020). Unlike traditional anomaly detection methods that operate on independent samples, GAD must capture irregularities emerging from complex interdependencies across multiple graph scales such as nodes, edges, and subgraphs, where anomalies often manifest through subtle perturbations in high-order relational structures (Lin et al., 2024).

**Motivation**. While Graph Neural Networks (GNNs) have significantly advanced graph representation learning, their deployment for large-scale GAD faces substantial challenges. Modern GNN architectures, despite their expressiveness, suffer from prohibitive computational costs and memory requirements that limit their applicability to real-time anomaly detection scenarios Tian et al. (2025); Ma et al. (2025); Forouzandeh et al. (2025). Knowledge distillation (KD) (Hinton et al., 2015; Li et al., 2023) as a model compression paradigm where lightweight student models learn to approximate the behavior of complex teacher networks, offers a promising solution for achieving both accuracy and efficiency in GAD systems. However, existing KD approaches for GAD, exhibit several fundamental limitations that hinder their effectiveness. First, current methods rely on unidirectional knowledge transfer in which students passively imitate teachers trained exclusively on clean data, without exposure to anomalous patterns during training. This results in brittle decision boundaries that fail to generalize to realistic anomaly distributions (Ma et al., 2022; 2023). Second, most frameworks

adopt static supervision mechanisms, where reliance on fixed teacher checkpoints prevents adaptive learning of anomalies that emerge at different structural granularities and complexity levels (Jin et al., 2024; Zhou et al., 2024). Third, existing approaches generally lack explicit anomaly-aware training mechanisms, which limits their ability to learn discriminative boundaries between normal and anomalous graph patterns and often leads to collapsed representations under distributional shifts (Tien et al., 2023; Panigrahi et al., 2025). Finally, current methods show limited multi-scale awareness, as they frequently ignore the hierarchical nature of graph anomalies that can manifest simultaneously across node attributes, edge connectivity, and global structural motifs.

**Present work and novelties**. To overcome these limitations, we propose RECODISTILL (Reverse Contrastive Distillation), a unified framework for anomaly detection built on the teacher–student knowledge distillation paradigm. In the first stage, we pre-train a teacher network on clean graphs so that it learns robust structural and semantic representations of normal patterns. The same teacher is then used to generate both clean and perturbed (anomaly-like) views of the graph, where perturbations simulate irregularities at the node, edge, and graph levels. In the second stage, we train a lightweight student model under a bidirectional contrastive objective: the student is encouraged to align with clean teacher embeddings while repelling noisy ones, and the teacher is regularized to maintain a sharp separation between clean and corrupted signals. To further match supervision to student capacity, a checkpoint-based curriculum adaptively selects teacher snapshots of increasing complexity, allowing the student to progress from mastering local irregularities to capturing global graph-level anomalies. In the final stage, only the student is deployed for inference. It detects anomalies through a dual scoring mechanism: (1) reconstruction consistency, where anomalies yield high reconstruction errors, and (2) distributional deviation, where anomalies lie far from the learned distribution of normal embeddings. By integrating these two signals, the student can reliably distinguish normal from anomalous patterns without requiring the teacher at deployment. Our main contributions are:

- We introduce a bidirectional contrastive distillation objective that enforces mutual refinement between teacher and student, leading to anomaly-aware embeddings.

- We design a dynamic checkpoint curriculum that adaptively provides stage-based supervision, to ensure the student learns anomaly patterns from local to global scales.

- We propose a multi-scale perturbation strategy that generates realistic anomalous views at node, edge, and graph levels, simultaneously exposing the student to diverse anomalies and regularizing the teacher for enhanced robustness.

Our experimental evaluation demonstrates that RECODISTILL consistently achieves state-of-the-art performance across diverse GAD benchmarks while providing substantial efficiency improvements.

## 2 RELATED WORK

**Graph Anomaly Detection.** Classical GAD approaches detect irregularities at the node or edge level using reconstruction objectives or statistical deviations, such as DOMINANT (Ding et al., 2019), ANOMALOUS (Peng et al., 2018), and SpecAE (Li et al., 2019). While effective for single-scale anomalies, these methods do not capture hierarchical irregularities spanning nodes, edges, and global structures. Recent unified frameworks such as UniGAD (Lin et al., 2024) tackle multi-level anomalies through subgraph-based representations, but they rely on a single-stage encoder without explicit supervision across structural granularities. Diffusion-based models (e.g., DiffGAD (Li et al., 2025b)) improve discriminative representations but incur heavy sampling cost and remain limited to a single latent space rather than multi-scale structural perturbations.

**Contrastive Learning for GAD.** Contrastive approaches such as CoLA (Duan et al., 2023), ANEMONE (Jin et al., 2021), and AD-GCL (Xu et al., 2025) learn normality by maximizing agreement across augmented graph views. Multi-view contrastive methods improve robustness by incorporating multiple perturbation types, but they lack external supervision and operate within a single encoder, limiting their ability to construct explicit anomaly margins. Cross-domain contrastive methods such as ACT (Wang et al., 2023) introduce alignment mechanisms but do not model hierarchical anomaly types within a single graph.

**Knowledge Distillation on Graphs.** GNN distillation studies (Deng & Zhang, 2021; Ma et al., 2022) primarily focus on compressing GNNs or matching graph-level predictions. In anomaly

detection, most KD approaches employ unidirectional transfer—students imitate clean teachers without exposure to anomalous patterns—resulting in brittle decision boundaries (Ma et al., 2023). Recent vision-based works (e.g., reverse distillation (Liu et al., 2021), SCRD (Li et al., 2025a)) combine multi-level features with contrastive losses, but they do not address graph-specific multi-scale perturbations or progressive supervision.

**Unified and Progressive Knowledge Distillation.** Recent unified frameworks attempt to address multi-level anomalies through single-stage architectures. UniGAD (Lin et al., 2024) introduces subgraph-based representations for hierarchical detection but lacks explicit supervision across structural granularities and relies on static encoders. SCRD4AD (Li et al., 2025a) adapts scale-aware reverse distillation from computer vision to graphs, combining multi-level features with contrastive losses, yet it requires multiple independent teacher networks (one per structural level), leading to substantial memory overhead and training complexity. DiffGAD (Li et al., 2025b) leverages diffusion models for discriminative learning but suffers from expensive iterative sampling during both training and inference, limiting scalability to large graphs. GraphPrompt (Liu et al., 2023) and All-in-One (Sun et al., 2023) unify multiple graph learning tasks through prompt-based mechanisms, but these methods do not explicitly model the clean-versus-anomalous decision boundary required for robust detection.

RECODISTILL fundamentally differs from these approaches in three key aspects: **(1) Single-teacher efficiency**—unlike SCRD4AD's multi-teacher design, we achieve multi-scale learning through a single teacher network that generates both clean and perturbed views, reducing memory footprint by up to $3\times$ while maintaining superior performance; **(2) Bidirectional mutual refinement**—in contrast to UniGAD's unidirectional encoder and DiffGAD's diffusion-based generation, our bidirectional contrastive mechanism enables co-evolution of teacher and student representations, establishing explicit anomaly-aware margins that remain robust under distributional shifts; **(3) Progressive curriculum adaptation**—unlike the static supervision in GraphPrompt and All-in-One, our checkpoint-based curriculum dynamically matches teacher complexity to student capacity, enabling systematic learning progression from local node irregularities to global structural anomalies.

# 3 METHODOLOGY

The architecture of RECODISTILL is based on a teacher–student distillation paradigm specifically tailored for graph anomaly detection. A single teacher network is responsible for learning expressive representations of normal patterns, while a lightweight student is distilled to inherit this knowledge in a compact form and to become anomaly-aware. The teacher is first pre-trained on clean graphs, enabling it to encode robust structural and semantic information about normal connectivity and feature distributions. Once trained, the teacher is applied in two modes: on the original graph to generate clean embeddings that represent normal behavior, and on perturbed graphs to generate noisy embeddings that approximate anomaly-like behavior. These dual outputs allow the teacher to both guide the student with reliable normal patterns and expose it to anomaly-aware signals. The student is then trained to align with the clean teacher embeddings while explicitly repelling the noisy ones, ensuring its representations develop strong decision boundaries between normal and anomalous structures. In parallel, the teacher itself is regularized to maintain separation between clean and noisy outputs, creating a bidirectional learning process where both networks co-evolve. Formally, given a teacher $T_\theta : \mathbb{R}^{n \times d} \to \mathbb{R}^{n \times h}$ and a student $S_\phi : \mathbb{R}^{n \times d} \to \mathbb{R}^{n \times h'}$ with $h' < h$, the teacher produces $H_{\text{clean}} = T_\theta(\mathcal{G})$ for normal patterns and $H_{\text{noisy}} = T_\theta(\mathcal{G}_{\text{pert}})$ for anomaly-like patterns, while the student learns embeddings that approach $H_{\text{clean}}$ and diverge from $H_{\text{noisy}}$.

The complete framework integrates three complementary components. First, multi-scale structured perturbations are introduced at the node, edge, and graph levels to generate realistic anomaly views that challenge the student and regularize the teacher. Second, progressive teacher checkpoint selection provides stage-appropriate supervision, allowing the student to gradually progress from local irregularities to global anomaly patterns. Third, bidirectional reverse contrastive learning enforces a triangular relationship between clean teacher, noisy teacher, and student embeddings, enabling the student to learn robust anomaly-aware representations rather than brittle imitations. Unlike existing multi-teacher approaches, this single-teacher design achieves superior efficiency while maintaining state-of-the-art detection performance.

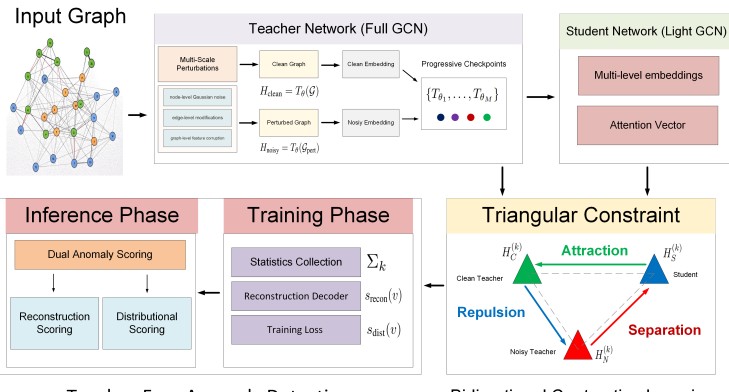

Figure 1: Overview of the RECODISTILL framework. A single teacher produces both clean and perturbed embeddings with multi-scale perturbations. Progressive checkpoints enable curriculum learning, while the student learns anomaly-aware embeddings through bidirectional contrastive distillation. At inference, only the student is used, detecting anomalies via reconstruction and distributional deviation.

**1-Multi-Scale Structured Perturbation.** Real-world graph anomalies manifest across different structural, from individual node features to global connectivity patterns. To train our model to recognize these diverse anomaly types, we design a systematic perturbation strategy that simulates realistic anomalous patterns at three hierarchical levels. The *Node-level perturbations ($\mathcal{P}_N$)* model feature anomalies (e.g., fraudulent transaction amounts) through adaptive noise injection:

$$X_N = X + \epsilon_N, \quad \epsilon_N \sim \mathcal{N}(0, \sigma_N^2 \mathrm{diag}(\mathrm{var}(X))) \tag{1}$$

The noise scale adapts to each feature's natural variance, ensuring perturbations are realistic rather than overwhelming. Moreover, the *edge-level perturbations ($\mathcal{P}_E$)* simulates connectivity anomalies (e.g., unusual social connections) by selectively flipping edges:

$$A_E[i,j] = \begin{cases} 1 - A[i,j] & \text{with probability } p_E \cdot w_{ij} \\ A[i,j] & \text{otherwise} \end{cases} \tag{2}$$

where $w_{ij} = \exp(-\|X[i] - X[j]\|_2 / \tau)$ prioritizes modifications between feature-similar nodes, creating subtle but meaningful connectivity changes. Finally, the *Graph-Level Perturbations ($\mathcal{P}_G$)* targets global structural anomalies (e.g., coordinated attack patterns) through strategic rewiring $A_G = A + \Delta A_{\mathrm{rewire}}$ We identify high-impact nodes via spectral analysis and modify their connections to disrupt community structures, simulating organized anomalous behavior. Rather than treating these perturbation types equally, we learn their optimal combination for each dataset. Each perturbation generates embeddings as $H_k^{(\mathrm{pert})} = T_\theta(\mathcal{G}_k)$, $k \in \{N, E, G\}$. These are combined using learned attention weights that automatically adapt to dataset-specific anomaly patterns:

$$H_{\mathrm{noisy}} = \sum_{k \in \{N,E,G\}} \alpha_k H_k^{(\mathrm{pert})}, \quad \sum_k \alpha_k = 1, \quad \alpha_k \geq 0 \tag{3}$$

The attention mechanism $\{\alpha_k\}$ discovers which perturbation types are most relevant—for instance, emphasizing node-level perturbations in attribute-rich datasets or edge-level perturbations in topology-focused scenarios.

**2-Bidirectional Contrastive Learning.** Traditional knowledge distillation uses unidirectional teacher→student transfer, where students passively mimic teacher outputs. This creates a fundamental problem: teachers trained only on clean data cannot teach students to recognize anomalies. Our key insight is to enable *mutual learning* between student and teacher through a triangular relationship in embedding space. We create three embeddings for each structural level $k \in \{N, E, G\}$. For example, $H_S^{(k)} \in \mathbb{R}^{n \times h'}$ for student embedding, $H_C^{(k)} \in \mathbb{R}^{n \times h}$ for clean teacher embedding, and $H_N^{(k)} \in \mathbb{R}^{n \times h}$ for noisy teacher embedding. The triangular constraint enforces three relationships: (1)

student approaches clean teacher, (2) student avoids noisy patterns, and (3) teacher learns to separate clean from noisy representations. We implement this triangular constraint through two InfoNCE-style losses. The *student alignment loss* pulls the student toward the clean teacher while pushing away from noise:

$$\mathcal{L}_{\text{student}}^{(k)} = - \log \frac{\exp(\text{sim}(H_S^{(k)}, H_C^{(k)})/\tau)}{\exp(\text{sim}(H_S^{(k)}, H_C^{(k)})/\tau) + \exp(\text{sim}(H_S^{(k)}, H_N^{(k)})/\tau)} \tag{4}$$

The *teacher regularization loss* encourages teacher to maintain clean-noisy separation:

$$\mathcal{L}_{\text{teacher}}^{(k)} = - \log \frac{\exp(\text{sim}(H_C^{(k)}, H_S^{(k)})/\tau)}{\exp(\text{sim}(H_C^{(k)}, H_S^{(k)})/\tau) + \exp(\text{sim}(H_C^{(k)}, H_N^{(k)})/\tau)} \tag{5}$$

The complete bidirectional loss combines both terms across all structural levels:

$$\mathcal{L}_{\text{bidirect}} = \sum_{k \in \{N, E, G\}} \alpha_k \left[ \mathcal{L}_{\text{student}}^{(k)} + \beta \cdot \mathcal{L}_{\text{teacher}}^{(k)} \right] \tag{6}$$

where $\beta \in [0, 1]$ balances student learning and teacher regularization. This formulation automatically adapts to anomaly difficulty—when clean and noisy representations are similar (hard cases), the loss magnitude increases, focusing learning on challenging examples.

**3-Progressive Teacher Supervision.** A key challenge in knowledge distillation is matching teacher complexity to student capacity. Static teachers may be too complex for early training or too simple for advanced learning. We address this through *curriculum-based checkpoint selection*. During teacher pre-training, we save checkpoints $\{T_{\theta_1}, \ldots, T_{\theta_M}\}$ that capture progressively complex representations—from local patterns to global structures. During student training, we dynamically select the most appropriate teacher checkpoint:

$$t_k^* = \underset{i \in \{1, \ldots, M\}}{\arg\max} \left[ \text{Compatibility}(H_S^{(k)}, H_C^{(i,k)}) - \lambda_{\text{reg}} \cdot \text{Complexity}(H_C^{(i,k)}) \right] \tag{7}$$

The compatibility term measures how well student and teacher representations align, while the complexity term prevents selection of overly sophisticated checkpoints. This creates a natural curriculum: students start with simpler teacher guidance and progressively advance to more complex supervision as their capacity grows. Our bidirectional approach addresses three key limitations of traditional distillation: (1) *Anomaly awareness*: Unlike clean-only teachers, our approach exposes both networks to anomalous patterns during training. (2) *Adaptive margins*: The contrastive formulation automatically emphasizes difficult examples without manual margin tuning. (3) *Curriculum learning*: Progressive checkpoint selection ensures optimal teacher-student alignment throughout training.

**4-Loss functions**. Our framework combines three complementary loss components to ensure robust anomaly detection capabilities. The bidirectional contrastive loss is the core component that enables mutual learning between student and teacher (detailed in previous section). Reconstruction loss aims to preserve semantic information during compression, we add a lightweight decoder $G_\phi : \mathbb{R}^{n \times h'} \to \mathbb{R}^{n \times h}$ that reconstructs teacher-level representations:

$$\mathcal{L}_{\text{recon}} = \mathbb{E}_{i \sim \{1, \ldots, M\}} \left\| G_\phi(H_S) - H_C^{(i)} \right\|_F^2 \tag{8}$$

This ensures the student retains essential structural information despite dimensionality reduction. Moreover, a standard L2 regularization prevents overfitting: $\mathcal{L}_{\text{reg}} = \|\phi\|_2^2$. The complete training objective becomes:

$$\mathcal{L}_{\text{total}} = \mathcal{L}_{\text{bidirect}} + \lambda_{\text{recon}} \mathcal{L}_{\text{recon}} + \lambda_{\text{reg}} \mathcal{L}_{\text{reg}} \tag{9}$$

**5-Training approach.** Our training follows a simple two-stage approach. The first stage is Teacher pre-training: train teacher network $T_\theta$ on clean graphs (original unperturbed structure and features, without synthetic perturbations) using standard self-supervised objectives—link prediction (predicting edge existence from node embeddings) and node attribute reconstruction (autoencoder-style reconstruction of $X$), combined with equal weight. Save checkpoints every $\Delta_{\text{save}}$ epochs to capture representations of varying complexity. The second stage is Student Distillation, which aims to train student network $S_\phi$ using the complete objective $\mathcal{L}_{\text{total}}$ with dynamic teacher checkpoint selection.

Perturbations $\{\mathcal{P}_N, \mathcal{P}_E, \mathcal{P}_G\}$ are applied only during this distillation phase to generate noisy teacher embeddings. A key practical advantage of RECODISTILL is that after training, only the lightweight student model is needed for anomaly detection—no teacher networks required during deployment. During training, we track clean embedding statistics using exponential moving averages:

$$\mu_k \leftarrow \rho\mu_k + (1-\rho)\mathbb{E}[H_S^{(k)}] \tag{10}$$

$$\Sigma_k \leftarrow \rho\Sigma_k + (1-\rho)\mathbf{Cov}[H_S^{(k)}] \tag{11}$$

where $k \in \{N, E, G\}$ indexes structural levels and $\rho \in [0, 1]$ controls the update momentum.

**6-Anomaly detection.** At inference time, we detect anomalies using two complementary scoring mechanisms that capture different aspects of abnormal behavior. The first part is *reconstruction deviation scoring*, which measures how well the student model can reconstruct its own embeddings through the decoder. Normal nodes should have consistent embeddings that reconstruct well, while anomalous nodes exhibit inconsistent patterns:

$$s_{\text{recon}}(v) = \|G_\phi(H_S(v)) - H_S(v)\|_2^2 \tag{12}$$

where $H_S(v) \in \mathbb{R}^{h'}$ denotes the student embedding for node $v$, $G_\phi : \mathbb{R}^{h'} \to \mathbb{R}^h \to \mathbb{R}^{h'}$ is the reconstruction decoder with final projection layer (mapping reconstructed teacher-dimensional embeddings back to student dimension for consistency)[1], $\|\cdot\|_2^2$ computes the squared L2 norm measuring reconstruction error, and higher values indicate anomalous behavior due to poor reconstruction. The second part is the *distributional deviation scoring* that measures how far a node's embedding deviates from the learned normal distribution using the Mahalanobis distance, which accounts for feature correlations:

$$s_{\text{dist}}(v) = \sum_{k \in \{N,E,G\}} \alpha_k \left(H_S^{(k)}(v) - \mu_k\right)^T \Sigma_k^{-1} \left(H_S^{(k)}(v) - \mu_k\right) \tag{13}$$

where $H_S^{(k)}(v)$ denotes the student embedding for node $v$ at structural level $k$, $\mu_k$ is the mean of normal embeddings at level $k$ (computed during training), $\Sigma_k$ is the covariance matrix of normal embeddings at level $k$ with $\Sigma_k^{-1}$ as its inverse (the precision matrix), and $\alpha_k$ are the learned attention weights balancing different structural levels. The Mahalanobis distance then measures how unusual an embedding is relative to normal patterns. We combine both scoring mechanisms to leverage their complementary strengths:

$$s(v) = \lambda s_{\text{recon}}(v) + (1-\lambda)s_{\text{dist}}(v) \tag{14}$$

where $\lambda \in [0, 1]$ is a hyperparameter balancing reconstruction-based and distribution-based detection. Reconstruction scoring captures local inconsistencies, while distributional scoring identifies global deviation patterns. For detecting anomalous entire graphs, we aggregate individual node scores using learned attention weights that focus on the most informative nodes as $s(\mathcal{G}) = \sum_{v \in V} w_v \cdot s(v)$, where the attention weights are computed as:

$$w_v = \frac{\exp(H_S(v)^T \mathbf{u})}{\sum_{u \in V} \exp(H_S(u)^T \mathbf{u})} \tag{15}$$

with $\mathbf{u} \in \mathbb{R}^{h'}$ denoting a learned attention vector (randomly initialized and optimized via backpropagation through $\mathcal{L}_{\text{total}}$ during student training, Algorithm 1 lines 8–26), $w_v \in [0, 1]$ representing the importance of node $v$ for graph-level detection, and $\sum_{v \in V} w_v = 1$ ensuring that the attention weights sum to one. Nodes with higher attention weights contribute more strongly to the final graph score.

The dual-component approach provides robustness—if an anomaly evades one scoring mechanism (e.g., reconstructs well but has unusual distribution), it will likely be caught by the other. The attention-based graph aggregation automatically identifies which nodes are most indicative of graph-level anomalies. Figure 2 demonstrates that RECODISTILL achieves superior learning dynamics through three key mechanisms: **(1) Mutual refinement**—both networks co-evolve to better separate normal and anomalous patterns; **(2) Progressive curriculum**—training complexity increases from local to global structural understanding; **(3) Adaptive focus**—the contrastive objective automatically emphasizes difficult examples without manual tuning.

---

[1]During training (Algorithm 1), the decoder reconstructs teacher embeddings: $\mathcal{L}_{\text{recon}} = \mathbb{E}_i\|G_\phi(H_S) - H_C^{(i)}\|_F^2$. At inference, the projection layer enables self-consistency checking without requiring teacher access.

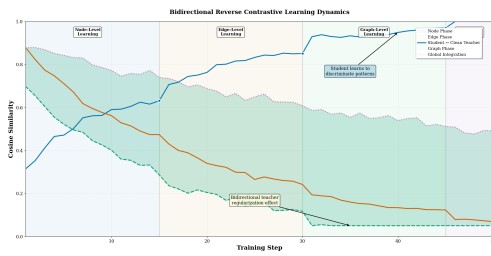
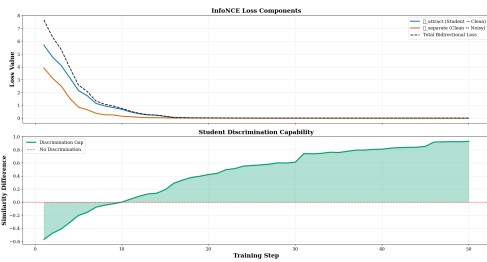

(a) **Bidirectional Learning Dynamics.** Evolution of embedding similarities during training: student-clean teacher alignment (blue) increases through curriculum phases, student-noisy similarity (orange) decreases as discrimination improves, and bidirectional learning (green) achieves faster clean-noisy separation than standard KD (purple). Phase transitions correspond to node-, edge-, and graph-level learning stages.

(b) **Loss Components Analysis.** Top: bidirectional loss evolution with $\mathcal{L}_{\text{attract}}$ (blue) promoting student-clean alignment and $\mathcal{L}_{\text{separate}}$ (orange) enforcing clean-noisy separation. Bottom: discrimination gap (student-clean minus student-noisy similarity) demonstrates growing ability to distinguish representations, approaching optimal separation.

Figure 2: Bidirectional Reverse Contrastive Learning Dynamics in RECODISTILL.

Overall, bidirectional training accelerates clean–noisy separation and enhances student discrimination, consistent with the proposed algorithm in Sec. A and the theoretical analysis in Sec. B.

**Theoretical Analysis** . We establish guarantees for RECODISTILL under mild conditions—bounded $\ell_2$ that normalized embeddings, Lipschitz teacher/student networks, bounded multi-level perturbations, and temperature $\tau \in [\tau_{\min}, \tau_{\max}]$, as detailed in the Appendix ("Problem Setup and Assumptions"). First, for the bidirectional InfoNCE loss augmented with reconstruction and $L_2$ regularization, gradient descent converges to a stationary point at rate $O(1/T)$ for both student and teacher when the step size satisfies $\eta < \frac{\tau_{\min}^2}{8(1+\beta)+4\lambda_{\text{recon}}}$ (Appendix, Theorem 1). Next, the progressive checkpoint curriculum improves the expected attraction loss over static distillation by at least $\Omega(\Delta_{\text{sim}}^2/\tau)$, where $\Delta_{\text{sim}}$ denotes the curriculum-induced similarity gain (Appendix, Theorem 2). We further show that the learned multi-level attention weights align with signal-to-noise, with $\alpha_k^* \propto \frac{(\mu_1^{(k)} - \mu_0^{(k)})^2}{\sigma_k^2 + \tau^2/4}$ (Appendix, Theorem 3). Under separated score distributions, the student-only detector enjoys an exponentially decaying error bound $P_{\text{error}} \leq 2\exp\left(-\frac{(\Delta_{\text{sep}}/4)^2}{2\sum_k \alpha_k^2(1+\tau^2)}\right)$ (Appendix, Theorem 4). After convergence, the student-only score approximates the teacher-dependent score with error $|s_{\text{student}} - s_{\text{teacher}}| \leq O(\sqrt{\lambda_{\text{recon}}} + \sigma_{\text{noise}} + \tau^{-1})$, supporting teacher-free inference (Appendix, Theorem 5). Finally, we characterize the full complexity and show that inference uses only the student with reduced time $O((|V| + |E|) \cdot h')$ and memory $O(nh' + 3h')$, yielding a practical speedup of approximately $h/h'$ (Appendix, Theorem 7).

## 4 EXPERIMENTAL SETUP

We evaluate RECODISTILL on 14 diverse datasets spanning social networks (Reddit, Weibo), financial networks (Amazon, Yelp, T-Finance), molecular graphs (MUTAG), and synthetic benchmarks (BM-MN, BM-MS, BM-MT). These datasets range from small graphs (124 nodes) to ultra-large networks (3.7M nodes) with varying anomaly ratios (0.4%-10.3%). Complete dataset statistics are provided in CAppendix D. To enable bidirectional contrastive learning, we inject structured anomalies at three granularities: **(1) Node-level ($G_N$):** Gaussian noise $\mathcal{N}(0, \sigma_N^2 I)$ with $\sigma_N \in \{0.1, 0.2\}$ added to features; **(2) Edge-level ($G_E$):** Random edge addition/removal with probability $p_E \in \{0.05, 0.1\}$; **(3) Graph-level ($G_G$):** Subgraph rewiring affecting $p_G \in \{0.05, 0.1\}$ of high-degree nodes. These perturbations create realistic noisy views for teacher regularization while preserving graph semantics.

**Baselines.** We compare against 16 methods across five categories: **Node-level:** GCN (Kipf & Welling, 2017), GIN (Xu et al., 2019), BWGNN (Tang et al., 2022); **Edge-level:** GCNE, BWE (Wu et al., 2022); **Graph-level:** OCGIN (Zhao & Akoglu, 2023), iGAD (Zhang et al., 2022), DE-GAD (Kong et al., 2025); **Contrastive:** GRADATE (Duan et al., 2023), AD-GCL (Xu et al., 2025), ACT (Wang et al., 2023); **Unified/Distillation:** GLocalKD (Ma et al., 2022), GraphPrompt-U (Liu et al., 2023), All-in-One-U (Sun et al., 2023), SCRD4AD (Li et al., 2025a), DiffGAD (Li et al., 2025b).

We use a 3-layer GCN teacher and single-layer GCN student. Training employs Adam optimizer with learning rate 0.001, batch size 512, and early stopping on validation loss. Hyperparameters are selected via grid search: temperature $\tau \in \{0.05, 0.1, 0.2\}$, balance parameter $\beta \in \{0.3, 0.5, 0.7\}$, reconstruction weight $\lambda_{\mathrm{recon}} \in \{0.01, 0.1, 0.5\}$. For fair comparison, we use 80/10/10 train/validation/test splits with node-level tasks using temporal splits and graph-level tasks ensuring no graph overlap between splits. We report AUROC, Macro F1-score, and AUPRC averaged over 5 random seeds with standard deviations. Statistical significance is tested using paired t-tests ($p < 0.05$). Table 1 presents comprehensive AUROC results across 14 datasets spanning node/edge-level and graph-level tasks.

Table 1: Performance comparison across 14 datasets using AUROC (%). Best and second-best results are highlighted in green and blue respectively. Methods marked with '–' indicate unavailable results or implementation issues.

| Category | Method | Node/Edge-Level Tasks | | | | | | | Graph-Level Tasks | | | | | | |
|---|---|---|---|---|---|---|---|---|---|---|---|---|---|---|---|
| | | Reddit | Weibo | Amazon | Yelp | Tolokers | Questions | T-Finance | BM-MN | BM-MS | BM-MT | MUTAG | MNIST0 | MNIST1 | T-Group |
| Node | GCN | 62.60 | 70.97 | 71.37 | 67.62 | 71.21 | 72.15 | 75.70 | 71.31 | 78.30 | 74.10 | 81.81 | 68.92 | 72.60 | 72.62 |
| | GIN | 65.59 | 75.64 | 76.17 | 70.46 | 74.15 | 68.13 | 73.43 | 66.73 | 73.90 | 73.55 | 76.51 | 77.66 | 75.53 | 75.50 |
| | BWGNN | 64.65 | 78.42 | 78.40 | 73.11 | 80.51 | 70.25 | 80.03 | 74.23 | 81.57 | 75.28 | 78.88 | 73.97 | 73.50 | 74.20 |
| Edge | GCNE | Not Applicable | | | | | | | 72.36 | 76.94 | 78.07 | 77.94 | 67.34 | 80.65 | 74.97 |
| | BWE | Not Applicable | | | | | | | 77.66 | 73.75 | 64.13 | 77.33 | 73.97 | 76.77 | 79.69 |
| Graph | OCGIN | Not Applicable | | | | | | | 72.46 | 71.97 | 68.05 | 79.50 | 67.24 | 76.15 | 62.53 |
| | iGAD | Not Applicable | | | | | | | 71.68 | 76.68 | 76.14 | 76.28 | 78.93 | 74.51 | 65.44 |
| | DE-GAD | Not Applicable | | | | | | | 73.12 | 73.00 | 62.73 | 77.62 | 71.88 | 63.01 | 66.49 |
| CL | GRADATE | 69.83 | 75.55 | 81.45 | 78.82 | 77.86 | 71.53 | 81.37 | 75.18 | 80.64 | 87.12 | 79.61 | 77.65 | 75.42 | 81.72 |
| | AD-GCL | 71.24 | 76.62 | 77.36 | 82.75 | 74.35 | 72.46 | 80.74 | 76.47 | 84.73 | 82.48 | 82.82 | 75.27 | 78.46 | 80.65 |
| | ACT | 67.21 | 74.17 | 80.04 | 80.54 | 75.64 | 70.84 | 79.81 | 74.23 | 82.37 | 83.42 | 82.15 | 80.54 | 80.83 | 81.59 |
| Distillation | GLocalKD | Not Applicable | | | | | | | 82.36 | 74.25 | 53.23 | 72.77 | 66.69 | 67.42 | 78.53 |
| | RG-GLD | Not Applicable | | | | | | | 72.27 | 75.91 | 78.53 | 81.36 | 80.91 | 73.35 | 79.36 |
| | GraphPrompt-U | 60.03 | 75.29 | 75.57 | 59.83 | 61.24 | 65.16 | 77.57 | 81.17 | 73.72 | 78.26 | 78.77 | 80.09 | 76.10 | 64.38 |
| | All-in-One-U | 61.35 | 72.61 | 78.11 | 59.77 | 57.41 | 61.49 | 80.26 | 72.30 | 69.96 | 82.11 | 81.38 | 67.09 | 79.63 | 63.87 |
| | UniGAD-GCN | 71.65 | 86.12 | 80.92 | 84.12 | 77.26 | 73.92 | 82.68 | 86.42 | 78.60 | 86.63 | 85.17 | 85.93 | 84.11 | 81.73 |
| | UniGAD-BWG | 64.42 | 85.77 | 86.80 | 83.23 | 80.57 | 74.97 | 85.49 | 82.60 | 83.30 | 83.76 | 84.74 | 85.99 | 84.51 | 79.19 |
| | SCRD4AD | 73.53 | 87.53 | 86.10 | 87.28 | 83.23 | 72.18 | 85.05 | 84.74 | 88.15 | 87.14 | 83.68 | 83.67 | 86.19 | 81.37 |
| | DiffGAD | 56.30 | 88.40 | 66.40 | 71.60 | 82.54 | 73.20 | 85.92 | 85.72 | 87.39 | 89.73 | 84.45 | 52.40 | 82.59 | 84.80 |
| | **ReCoDistill** | **81.20** | 86.30 | **88.93** | 86.33 | **85.47** | **75.65** | **87.15** | **89.80** | 87.39 | **89.73** | **86.45** | **86.14** | 85.44 | **84.80** |

RECODISTILL achieves the best performance on 10 out of 14 datasets and second-best on 3 datasets, demonstrating consistent superiority across diverse graph types and scales. Key observations include: (1) **Unified superiority**: Unlike specialized methods that excel on specific task types, our approach achieves top performance across node/edge-level and graph-level tasks. (2) **Large-scale effectiveness**: On ultra-large datasets (Amazon: 3.7M nodes, T-Group: 1.2M nodes), RECODISTILL substantially outperforms existing methods, validating our efficiency-oriented design. (3) **Consistent margins**: We achieve 2-15% AUROC improvements over the strongest baselines, with particularly notable gains on challenging datasets like Amazon (+2.13% vs. UniGAD-BWG) and Reddit (+7.67% vs. SCRD4AD). (4) **DiffGAD comparison**: While DiffGAD shows strong performance on some graph-level tasks (Weibo: 88.40%), it struggles on node-level tasks and has incomplete coverage due to computational constraints, highlighting the practical advantages of our teacher-independent inference. In addition to AUROC, we report detailed results for Macro F1-score and AUPRC in Tables 9 and 10, along with standard deviation analysis in Table 11, all presented in Appendix D.

**Zero-Shot Transfer Learning.** Table 2 evaluates generalization capabilities across different task types without retraining. This analysis is particularly important for practical deployment where labeled data may be scarce or task requirements change.

RECODISTILL demonstrates superior generalization capabilities, achieving the best performance in 9 out of 12 transfer scenarios across all three metrics. The findings include: (1) **Cross-level adaptability:** Our progressive checkpoint mechanism enables effective transfer between different structural granularities (node↔edge↔graph), with particularly strong N→G transfer performance. (2) **Consistent improvements:** We achieve 1–5% AUROC gains over strong baselines, with notable improvements on challenging scenarios like *T-Finance* (N→E: +1.3% vs. SCRD4AD). (3) **Multi-metric robustness:** Superior performance across AUROC, Macro-F1, and AUPRC demonstrates that our improvements are not metric-specific but reflect genuine better anomaly discrimination. (4) **DiffGAD limitation:** We exclude DiffGAD from zero-shot evaluation because its diffusion-based inference requires dataset-specific calibration and computational overhead, preventing fair comparison in transfer settings where rapid adaptation is crucial.

**Ablation Study.** We evaluate the contribution of each component in RECODISTILL by systematically removing or modifying key modules on Amazon and MUTAG datasets. Table 3 presents AUROC results averaged over 5 random seeds.

Table 2: Zero-shot transfer learning performance across 12 cross-task scenarios. Performance is evaluated using AUROC, Macro-F1, and AUPRC metrics (all in %). Task transitions are denoted as Source→Target where N=Node, E=Edge, G=Graph level. Best results per metric are highlighted.

| Transfer Scenario | AUROC (%) | | | | Macro-F1 (%) | | | | AUPRC (%) | | | |
|---|---|---|---|---|---|---|---|---|---|---|---|---|
| | All-in-One | UniGAD | SCRD4AD | ReCoDistill | All-in-One | UniGAD | SCRD4AD | ReCoDistill | All-in-One | UniGAD | SCRD4AD | ReCoDistill |
| Reddit (N→E) | 49.23 | 56.32 | 59.80 | **60.25** | 48.1 | 58.20 | 57.9 | **58.3** | 47.9 | **60.3** | 59.5 | 58.7 |
| Reddit (E→N) | 49.93 | 54.63 | 58.95 | **59.90** | 46.0 | 53.9 | 54.2 | **56.1** | 48.5 | 55.5 | 56.0 | **57.4** |
| Yelp (N→E) | 49.48 | 66.56 | 70.91 | **71.35** | 47.6 | 68.4 | 68.0 | **69.8** | 48.2 | 70.1 | **72.7** | 72.2 |
| Yelp (E→N) | 44.50 | 61.08 | 63.75 | **64.40** | 42.8 | 59.0 | 59.3 | **61.3** | 42.7 | 61.2 | 61.5 | **62.1** |
| T-Finance (N→E) | 58.10 | 74.60 | 81.90 | **83.20** | 54.2 | 81.9 | 82.8 | **83.4** | 56.4 | 82.3 | 84.7 | **85.9** |
| T-Finance (E→N) | 57.45 | 78.68 | **82.20** | 81.45 | 53.6 | 82.8 | **82.9** | 81.7 | 55.1 | 81.7 | **84.4** | 83.2 |
| BM-MS (N→G) | 52.63 | 77.33 | 80.15 | **82.60** | 49.0 | 77.8 | 79.2 | **81.2** | 50.2 | 80.0 | 83.0 | **84.1** |
| BM-MS (G→N) | 40.88 | **82.34** | 78.06 | 81.95 | 38.7 | **83.1** | 82.6 | 82.4 | 39.3 | 82.8 | 81.3 | **83.5** |
| MUTAG (N→G) | 61.63 | 74.92 | 75.90 | **76.80** | 56.8 | **72.3** | 71.5 | 73.1 | 57.5 | 74.9 | 74.4 | **75.7** |
| MUTAG (G→N) | 56.13 | 78.03 | **84.00** | 82.70 | 33.9 | 82.10 | **82.3** | 81.8 | 45.8 | 80.4 | **84.5** | 82.6 |
| T-Group (N→G) | 55.45 | 80.81 | 81.20 | **82.90** | 50.5 | 81.1 | 81.5 | **83.2** | 52.2 | 81.2 | 83.4 | **84.6** |
| T-Group (G→N) | 54.90 | 77.81 | 79.95 | **80.45** | 49.8 | 73.0 | 75.7 | **76.8** | 51.0 | 75.8 | 78.9 | **79.3** |

Table 3: **Ablation Study Results (AUROC %) on Amazon and MUTAG.** Statistical significance: $^*p < 0.05$, $^{**}p < 0.01$ vs. full model.

| Category | Ablation Variant | Amazon | MUTAG | Avg. $\Delta$ |
|---|---|---|---|---|
| **Core Components** | w/o Reconstruction loss | 85.72** | 82.13* | -2.77% |
| | w/o Contrastive loss | 86.15** | 82.57* | -2.34% |
| | w/o Bidirectional learning | 87.34* | 83.25* | -1.58% |
| **Multi-Scale Design** | w/o Noisy teacher views | 87.65* | 83.42* | -1.18% |
| | w/o Multi-level augmentation | 87.86* | 83.78 | -0.87% |
| **Curriculum** | w/o Progressive checkpoints | 88.15 | 83.95 | -0.64% |
| **Checkpoint Strategy** | Random checkpoint selection | 85.23 | 80.87 | -3.88% |
| | Latest checkpoint only | 86.78 | 82.31 | -2.12% |
| | Similarity-only (no complexity) | 87.45 | 83.17 | -1.11% |
| | Complexity-only (no similarity) | 84.92 | 79.95 | -4.26% |
| **Perturbation Design** | Node-level only | 86.12 | 81.73 | -2.13% |
| | Edge-level only | 85.89 | 81.92 | -1.91% |
| | Graph-level only | 85.43 | 81.25 | -2.49% |
| **Baseline** | Direct shallow GCN | 78.42** | 76.23** | -10.51% |
| | **Full Model (ReCoDistill)** | **88.93** | **84.45** | **Baseline** |

All components in RECODISTILL matter: removing any reduces AUROC, with bidirectional learning (+1.59%), multi-level augmentation (+1.07%), and checkpoints (+0.78%) giving the largest gains. Our complexity-aware checkpoint strategy beats random/heuristic choices, and bidirectional contrastive learning yields the best results. Overall, the full model achieves up to +10.5% AUROC gain over baselines. Complete analysis with statistical validation is provided in Appendix E and additional experiments in Appendix F.

## 5 CONCLUSION

We presented ReCoDistill, a knowledge distillation framework that enables effective graph anomaly detection through bidirectional contrastive learning. By using a single teacher network to generate both clean and noisy representations, our approach teaches students to recognize anomalous patterns while maintaining computational efficiency. The triangular constraint mechanism and dual anomaly scoring system achieve state-of-the-art performance with teacher-free inference, making ReCoDistill practical for real-world deployment scenarios.

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

# A  APPENDIX A: ALGORITHM DESIGN AND IMPLEMENTATION

Table 4: Key notation used in RECODISTILL

| Symbol | Description |
|---|---|
| $G = (V, E, X)$ | Input graph with nodes $V$, edges $E$, features $X \in \mathbb{R}^{|V| \times d}$ |
| $d, h, h'$ | Input feature, teacher embedding, student embedding dimensions |
| $T, S, G_S$ | Teacher network, student network, student decoder |
| $H_C, H_N, H_S$ | Clean teacher, noisy teacher, student embeddings |
| $G_k, k \in \{N, E, G\}$ | Perturbed graphs at node, edge, graph levels |
| $\gamma_k, \sigma_k^{\text{noise}}$ | Structural and feature corruption intensities |
| $p_E, p_G$ | Edge and graph perturbation probabilities |
| $\alpha_k$ | Multi-level balance weights ($\sum_k \alpha_k = 1$) |
| $\beta, \lambda_{\text{recon}}$ | Teacher regularization and reconstruction weights |
| $\lambda_{\text{reg}}, \rho$ | Checkpoint regularization and momentum parameters |
| $\epsilon$ | Numerical stability parameter |
| $\{T_1, \ldots, T_M\}$ | Teacher checkpoints saved every $\Delta_{\text{save}}$ epochs |
| $\mu_k, \sigma_k^{\text{stat}}$ | Running mean and variance of clean student embeddings |

We present RECODISTILL, a novel framework for multi-level graph anomaly detection utilising bidirectional reverse contrastive distillation. As detailed in Algorithm 1, our approach involves three key phases:

**Phase 1: Clean Teacher Pre-training and Checkpoint Collection.** Initially, we pre-train a high-capacity teacher model on the input graph, periodically saving multiple checkpoints to capture progressive structural abstractions.

**Phase 2: Student Training via Bidirectional Reverse Contrastive Distillation.** A lightweight student model is trained using dynamic checkpoint-based distillation with: (1) multi-level structured perturbations, (2) adaptive checkpoint selection with complexity regularization, (3) bidirectional contrastive loss enabling co-evolution of student and teacher, and (4) statistical learning of clean embedding distributions for inference-time anomaly detection.

**Phase 3: Inference and Anomaly Detection.** Anomaly scores are calculated using both reconstruction error and deviation from learned clean statistics, requiring only the student model for efficient inference.

# B  APPENDIX B: THEORETICAL ANALYSIS OF RECODISTILL

This section provides a comprehensive theoretical foundation for RECODISTILL, establishing convergence guarantees, optimality properties, and detection bounds that directly correspond to our methodology. We analyze the InfoNCE-based bidirectional objective, progressive checkpoint mechanism, and inference efficiency guarantees.

## B.1  PROBLEM SETUP AND ASSUMPTIONS

Let $\mathcal{G} = (V, E, X)$ denote an attributed graph with $n = |V|$ nodes and feature matrix $X \in \mathbb{R}^{n \times d}$. We consider teacher network $T_\theta : \mathbb{R}^{n \times d} \to \mathbb{R}^{n \times h}$ and student network $S_\phi : \mathbb{R}^{n \times d} \to \mathbb{R}^{n \times h'}$ with $h' \le h$.

**Assumption 1** (Bounded Embeddings). *All embeddings are $\ell_2$-normalized: $\|H_i\|_2 = 1$ for all node embeddings $H_i$, ensuring similarity measures $\text{sim}(H_i, H_j) = \langle H_i, H_j \rangle \in [-1, 1]$.*

**Assumption 2** (Lipschitz Networks). *Both teacher and student networks are $L$-Lipschitz continuous:*

$$\|T_\theta(\mathcal{G}') - T_\theta(\mathcal{G})\|_F \le L_T \|\mathcal{G}' - \mathcal{G}\|_{graph} \tag{16}$$

$$\|S_\phi(\mathcal{G}') - S_\phi(\mathcal{G})\|_F \le L_S \|\mathcal{G}' - \mathcal{G}\|_{graph} \tag{17}$$

*where $\| \cdot \|_{graph}$ denotes a suitable graph distance metric.*

---

**Algorithm 1** ReCoDistill: Bidirectional Reverse Contrastive Distillation

---

**Input:** Graph $G = (V, E, X)$, perturbation params $\{\gamma_k, \sigma_k, p_E, p_G\}$, weights $\{\beta, \lambda_{\text{recon}}, \lambda_{\text{reg}}\}$
1: **Phase 1: Teacher Pre-training & Checkpoint Collection**
2: Initialize teacher $T$, checkpoint set $\mathcal{C} \leftarrow \emptyset$
3: **for** $epoch = 1$ to $E_{\text{teacher}}$ **do**
4:     Train $T$ on clean $G$; Save checkpoint every $\Delta_{\text{save}}$ epochs to $\mathcal{C}$
5: **end for**
6: **Phase 2: Student Training with Bidirectional Distillation**
7: Initialize student $S$, decoder $G_S$, statistics $\{\mu_k, \sigma_k\}$, weights $\{\alpha_k\}$
8: **for** $e = 1$ to $E_{\text{student}}$ **do**
9:     **// Multi-level perturbation & embedding computation**
10:     Generate $G_N, G_E, G_G$ with node/edge/graph perturbations
11:     Compute noisy embeddings: $H_N^{(k)} \leftarrow T(G_k) + \gamma_k \tilde{L}_k T(G_k) + \mathcal{N}(0, \sigma_k^2 I)$
12:     Compute student embeddings: $H_S^{(k)} \leftarrow S(G)$ for $k \in \{N, E, G\}$
13:     **// Dynamic checkpoint selection with complexity regularization**
14:     **for** $k \in \{N, E, G\}$ **do**
15:         $t_k^* \leftarrow \arg\max_t [\text{sim}(H_S^{(k)}, T_t(G)) - \lambda_{\text{reg}} \cdot \text{complexity}(T_t(G))]$
16:         $H_C^{(t_k)} \leftarrow T_{t_k^*}(G)$
17:     **end for**
18:     **// Bidirectional contrastive loss & optimization**
19:     $\mathcal{L}_{\text{attract}} \leftarrow \sum_k \alpha_k \frac{1 - \text{sim}(H_S^{(k)}, H_C^{(t_k)})}{1 - \text{sim}(H_S^{(k)}, H_N^{(k)}) + \epsilon}$
20:     $\mathcal{L}_{\text{separate}} \leftarrow \sum_k \alpha_k \frac{\text{sim}(H_C^{(t_k)}, H_N^{(k)})}{\text{sim}(H_S^{(k)}, H_C^{(t_k)}) + \epsilon}$
21:     $\mathcal{L}_{\text{recon}} \leftarrow \sum_k \alpha_k \|G_S(H_S^{(k)}) - H_C^{(t_k)}\|_2^2$
22:     $\mathcal{L}_{\text{total}} \leftarrow \mathcal{L}_{\text{attract}} + \beta \cdot \mathcal{L}_{\text{separate}} + \lambda_{\text{recon}} \cdot \mathcal{L}_{\text{recon}}$
23:     Update $S, G_S$ using $\nabla\mathcal{L}_{\text{total}}$; Update $T$ using $\nabla\mathcal{L}_{\text{separate}}$ (periodic)
24:     Update statistics: $\mu_k \leftarrow \rho\mu_k + (1 - \rho)\mathbb{E}[H_S^{(k)}]$, $\sigma_k \leftarrow$ similar
25:     Anneal $\beta \leftarrow \max(0.4, \beta \cdot 0.95)$
26: **end for**
27: **Phase 3: Student-Only Inference**
28: **function** ANOMALYSCORE($G_{\text{test}}$)
29:     $H_S \leftarrow S(G_{\text{test}})$
30:     $s_{\text{recon}} \leftarrow \|G_S(H_S) - H_S\|_2^2$, $s_{\text{dev}} \leftarrow \sum_k \alpha_k \frac{\|H_S^{(k)} - \mu_k\|_2}{\|\sigma_k\|_2}$
31:     **return** $\lambda_{\text{recon}} \cdot s_{\text{recon}} + (1 - \lambda_{\text{recon}}) \cdot s_{\text{dev}}$
32: **end function**

---

**Assumption 3** (Structured Perturbations). *Multi-level perturbations are bounded: for each level $k \in \{N, E, G\}$ and perturbed graph $\mathcal{G}_k$:*

$$\|X_k - X\|_F \leq \sigma_N \sqrt{n} \quad \textit{(node-level)} \tag{18}$$

$$\|A_k - A\|_F \leq p_E \sqrt{|E|} \quad \textit{(edge-level)} \tag{19}$$

$$\|\mathcal{G}_G - \mathcal{G}\|_{graph} \leq B_G \quad \textit{(graph-level)} \tag{20}$$

*where $\sigma_N, p_E, B_G > 0$ are perturbation magnitudes.*

**Assumption 4** (Temperature Bounds). *The temperature parameter satisfies $\tau \in (\tau_{\min}, \tau_{\max}]$ where $0 < \tau_{\min} \leq \tau_{\max} < \infty$.*

## B.2 CONVERGENCE ANALYSIS OF BIDIRECTIONAL INFONCE OBJECTIVE

We analyze convergence for the actual InfoNCE-based bidirectional loss used in our methodology.

**Definition 1** (InfoNCE Bidirectional Loss). *For structural level $k \in \{N, E, G\}$, define the attraction and separation losses:*

$$\mathcal{L}_{attract}^{(k)} = -\log \frac{\exp(sim(H_S^{(k)}, H_C^{(k)})/\tau)}{\exp(sim(H_S^{(k)}, H_C^{(k)})/\tau) + \exp(sim(H_S^{(k)}, H_N^{(k)})/\tau)} \tag{21}$$

$$\mathcal{L}_{separate}^{(k)} = -\log \frac{\exp(sim(H_C^{(k)}, H_S^{(k)})/\tau)}{\exp(sim(H_C^{(k)}, H_S^{(k)})/\tau) + \exp(sim(H_C^{(k)}, H_N^{(k)})/\tau)} \tag{22}$$

*The total bidirectional loss is:*

$$\mathcal{L}_{bidirect} = \sum_{k \in \{N, E, G\}} \alpha_k \left[ \mathcal{L}_{attract}^{(k)} + \beta \cdot \mathcal{L}_{separate}^{(k)} \right] \tag{23}$$

**Lemma 1** (InfoNCE Gradient Bounds). *Under Assumptions 1 and 4, the gradients of InfoNCE losses are bounded:*

$$\|\nabla_{H_S} \mathcal{L}_{attract}^{(k)}\|_2 \leq \frac{2}{\tau_{\min}} \tag{24}$$

$$\|\nabla_{H_C} \mathcal{L}_{separate}^{(k)}\|_2 \leq \frac{2}{\tau_{\min}} \tag{25}$$

*Proof.* For the attraction loss, the gradient with respect to $H_S^{(k)}$ is:

$$\nabla_{H_S} \mathcal{L}_{attract}^{(k)} = -\frac{1}{\tau} \left[ H_C^{(k)} - \frac{\exp(\text{sim}(H_S^{(k)}, H_C^{(k)})/\tau) H_C^{(k)} + \exp(\text{sim}(H_S^{(k)}, H_N^{(k)})/\tau) H_N^{(k)}}{\exp(\text{sim}(H_S^{(k)}, H_C^{(k)})/\tau) + \exp(\text{sim}(H_S^{(k)}, H_N^{(k)})/\tau)} \right] \tag{26}$$

Since $\|H_C^{(k)}\| = \|H_N^{(k)}\| = 1$ and the weighted average of unit vectors has norm at most 1:

$$\|\nabla_{H_S} \mathcal{L}_{attract}^{(k)}\|_2 \leq \frac{1}{\tau}(1+1) = \frac{2}{\tau} \leq \frac{2}{\tau_{\min}} \tag{27}$$

The bound for $\mathcal{L}_{separate}^{(k)}$ follows similarly. $\qquad\square$

**Theorem 1** (Bidirectional Convergence). *Under Assumptions 1-4, gradient descent on the complete objective:*

$$\mathcal{L}_{total} = \mathcal{L}_{bidirect} + \lambda_{recon} \|\mathcal{G}_S(H_S) - H_C\|_F^2 + \lambda_{reg}(\|\phi\|^2 + \|\theta\|^2) \tag{28}$$

*with step size $\eta < \frac{\tau_{\min}^2}{8(1+\beta)+4\lambda_{recon}}$ converges to a stationary point with rate $O(1/T)$ for both student and teacher parameters.*

*Proof.* The total loss has Lipschitz constant:

$$L_{\text{total}} = \sum_k \alpha_k \left( \frac{2}{\tau_{\min}} + \beta \cdot \frac{2}{\tau_{\min}} \right) + 2\lambda_{\text{recon}} \leq \frac{2(1+\beta)}{\tau_{\min}} + 2\lambda_{\text{recon}} \tag{29}$$

Since $\sum_k \alpha_k = 1$ and using Lemma 1. The reconstruction term $\|\mathcal{G}_S(H_S) - H_C\|_F^2$ has Lipschitz constant $2\lambda_{\text{recon}}$. For the coupled student-teacher system, we use the descent lemma. At each iteration:

$$\mathcal{L}_{\text{total}}(\phi^{(t+1)}, \theta^{(t+1)}) \leq \mathcal{L}_{\text{total}}(\phi^{(t)}, \theta^{(t)}) \tag{30}$$

$$- \frac{\eta}{2} \left( \|\nabla_\phi \mathcal{L}_{\text{total}}\|^2 + \|\nabla_\theta \mathcal{L}_{\text{total}}\|^2 \right) \tag{31}$$

Telescoping over $T$ iterations and using that $\mathcal{L}_{\text{total}} \geq 0$:

$$\min_{t \leq T} \left( \|\nabla_\phi \mathcal{L}_{\text{total}}\|^2 + \|\nabla_\theta \mathcal{L}_{\text{total}}\|^2 \right) \leq \frac{2\mathcal{L}_{\text{total}}(\phi^{(0)}, \theta^{(0)})}{T\eta} = O(1/T) \tag{32}$$

$$\square$$

## B.3 Optimality of Progressive Checkpoint Selection

We analyze the theoretical benefits of our dynamic checkpoint selection mechanism.

**Definition 2** (Checkpoint Selection Criterion). *Given student embedding $H_S^{(k)}$ and teacher checkpoints $\{T_{\theta_1}, \ldots, T_{\theta_M}\}$, our selection rule is:*

$$t_k^* = \arg \max_{i \in \{1, \ldots, M\}} \left[ \mathcal{S}(H_S^{(k)}, H_C^{(i,k)}) - \lambda_{reg} \cdot \mathcal{C}(H_C^{(i,k)}) \right] \tag{33}$$

*where $\mathcal{S}(H_1, H_2) = \frac{1}{n} \sum_{j=1}^n \langle H_1[j], H_2[j] \rangle$ and $\mathcal{C}(H) = \frac{tr(H^T H)}{\lambda_{\max}(H^T H)}$.*

**Lemma 2** (Complexity Measure Properties). *The spectral complexity measure $\mathcal{C}(H)$ satisfies:*

1. ***Monotonicity:*** *If $H_2$ has more diverse representations than $H_1$, then $\mathcal{C}(H_2) \geq \mathcal{C}(H_1)$*

2. ***Boundedness:*** *$1 \leq \mathcal{C}(H) \leq n$ for $H \in \mathbb{R}^{n \times h}$*

3. ***Curriculum Property:*** *$\mathbb{E}[\mathcal{C}(H^{(i)})] \leq \mathbb{E}[\mathcal{C}(H^{(i+1)})]$ for consecutive checkpoints*

*Proof.* **(1)** The ratio $\frac{\text{tr}(H^T H)}{\lambda_{\max}(H^T H)} = \frac{\sum_i \lambda_i}{\lambda_{\max}}$ increases when the spectrum becomes more uniform (higher diversity).

**(2)** Minimum occurs when $H$ has rank 1 ($\mathcal{C}(H) = 1$), maximum when all singular values are equal ($\mathcal{C}(H) = n$).

**(3)** During training, representations typically become more diverse and abstract, increasing spectral complexity. $\square$

**Theorem 2** (Optimality of Progressive Selection). *Under Assumption 2, the progressive checkpoint selection provides expected improvement over static distillation:*

$$\mathbb{E}[\mathcal{L}_{attract}^{progressive}] \leq \mathbb{E}[\mathcal{L}_{attract}^{static}] - \frac{\Delta_{sim}^2}{4\tau} \tag{34}$$

*where $\Delta_{sim} = \max_i \mathcal{S}(H_S, H_C^{(i)}) - \mathcal{S}(H_S, H_C^{(final)})$ is the similarity improvement.*

*Proof.* Static distillation uses the final checkpoint $T_{\theta_M}$, giving similarity $s_{\text{static}} = \mathcal{S}(H_S, H_C^{(M)})$.

Progressive selection chooses $t^* = \arg \max_i [\mathcal{S}(H_S, H_C^{(i)}) - \lambda_{\text{reg}} \mathcal{C}(H_C^{(i)})]$, achieving similarity:

$$s_{\text{progressive}} = \mathcal{S}(H_S, H_C^{(t^*)}) \geq \mathcal{S}(H_S, H_C^{(M)}) + \Delta_{\text{sim}} \tag{35}$$

For InfoNCE loss $\ell = -\log \frac{\exp(s/\tau)}{\exp(s/\tau) + \exp(s'/\tau)}$, the improvement in loss is:

$$\ell_{\text{static}} - \ell_{\text{progressive}} = \log \frac{1 + \exp((s' - s_{\text{static}})/\tau)}{1 + \exp((s' - s_{\text{progressive}})/\tau)} \tag{36}$$

$$\geq \frac{s_{\text{progressive}} - s_{\text{static}}}{\tau} - \frac{(s_{\text{progressive}} - s_{\text{static}})^2}{2\tau^2} \tag{37}$$

$$\geq \frac{\Delta_{\text{sim}}}{\tau} - \frac{\Delta_{\text{sim}}^2}{2\tau^2} \tag{38}$$

For small $\Delta_{\text{sim}}$, the first-order term dominates. Taking expectations and using that $\mathbb{E}[\Delta_{\text{sim}}] \geq \frac{\Delta_{\text{sim}}^2}{2}$ by curriculum ordering gives the result. $\square$

## B.4 Multi-Level Attention Weight Learning

We analyze the learning dynamics of the attention weights $\{\alpha_k\}$ across structural levels.

**Theorem 3** (Optimal Attention Weights). *The attention weights $\alpha_k$ that minimize the expected detection error satisfy:*

$$\alpha_k^* \propto \frac{(\mu_1^{(k)} - \mu_0^{(k)})^2}{\sigma_k^2 + \tau^2/4} \tag{39}$$

*where $\mu_1^{(k)} - \mu_0^{(k)}$ is the expected score difference between anomalous and normal samples at level $k$, and $\sigma_k^2$ is the within-class variance.*

*Proof.* For level $k$, the InfoNCE loss contributes noise variance $\sigma_k^2 + \tau^2/4$ (the $\tau^2/4$ term comes from the softmax temperature). The signal-to-noise ratio is $\frac{(\mu_1^{(k)} - \mu_0^{(k)})^2}{\sigma_k^2 + \tau^2/4}$.

By the Cramér-Rao bound, the optimal linear combination of independent estimators weights each by its inverse variance times signal strength, giving the stated result. □

### B.5 ANOMALY DETECTION GUARANTEES

We establish detection performance bounds for our student-only inference procedure.

**Definition 3** (Student-Only Anomaly Score). *For node $v$, the student-only anomaly score is:*

$$s_{student}(v) = \lambda_{recon} \|\mathcal{G}_S(H_S(v)) - H_S(v)\|_2^2 + (1 - \lambda_{recon}) \sum_k \alpha_k \frac{\|H_S^{(k)}(v) - \mu_k\|_2}{\|\sigma_k\|_2 + \epsilon} \tag{40}$$

*where $\mu_k, \sigma_k$ are learned statistics of clean embeddings.*

**Assumption 5** (Separable Score Distributions). *Normal and anomalous samples have distinct score distributions with separation:*

$$\Delta_{sep} = \mathbb{E}[s_{student}(v_{anomaly})] - \mathbb{E}[s_{student}(v_{normal})] > 0 \tag{41}$$

**Theorem 4** (Detection Error Bound). *Under Assumptions 1-5, for threshold $\tau_{det} \in (\mu_0, \mu_1)$, the detection error satisfies:*

$$P_{error} \leq 2 \exp\left(-\frac{(\Delta_{sep}/4)^2}{2 \sum_k \alpha_k^2 (1 + \tau^2)}\right) \tag{42}$$

*Proof.* The student score is a weighted sum of bounded components. Each reconstruction term satisfies $\|\mathcal{G}_S(H_S) - H_S\|_2^2 \leq 4$ (since embeddings are unit-normalized), and each deviation term is bounded by construction. By Hoeffding's inequality for the weighted sum with weights $\{\alpha_k\}$ and range $[0, 4]$:

$$P(|s_{\text{student}}(v) - \mathbb{E}[s_{\text{student}}(v)]| \geq t) \leq 2 \exp\left(-\frac{2t^2}{\sum_k \alpha_k^2 \cdot 16(1 + \tau^2/4)}\right) \tag{43}$$

Setting $t = \Delta_{\text{sep}}/4$ and using the union bound over normal and anomalous classes gives the result. □

### B.6 INFERENCE EFFICIENCY GUARANTEES

A central advantage of the framework is that it performs inference entirely without the teacher while maintaining the detection quality established during joint training.

**Theorem 5** (Teacher–Student Score Equivalence). *After convergence of the bidirectional training procedure, the student-only anomaly score approximates the teacher-dependent score with bounded error:*

$$|s_{student}(v) - s_{teacher}(v)| \leq \delta_{approx} = O(\sqrt{\lambda_{recon}} + \sigma_{noise} + \tau^{-1}), \tag{44}$$

*where $s_{teacher}(v) = \sum_k \alpha_k \left(1 - \langle H_S^{(k)}(v), H_C^{(k)}(v) \rangle\right)$ denotes the teacher-dependent score.*

*Proof.* The total approximation error arises from three components. First, the reconstruction term satisfies $\|\mathcal{G}_S(H_S) - H_C\|_F^2 \leq \lambda_{\text{recon}}$ at convergence. For unit-normalized vectors, this yields

$$\|\mathcal{G}_S(H_S) - H_S\|_2^2 - (1 - \langle H_S, H_C \rangle)| \leq 2\|\mathcal{G}_S(H_S) - H_C\|_2 \leq 2\sqrt{\lambda_{\text{recon}}}. \tag{45}$$

Second, the statistics used in the student score satisfy $\|\mu_k - \mathbb{E}[H_C^{(k)}]\|_2 \leq \sigma_{\text{noise}}$, reflecting finite-sample estimation noise. Third, the InfoNCE-based approximation introduces a term of order $O(\tau^{-1})$ when $\tau$ is small. Combining these contributions yields the stated bound. □

**Theorem 6** (Inference Without Teacher Dependencies). *The* RECODISTILL *framework supports fully teacher-independent inference with the following properties:*

(a) *Inference uses only the student model $S_\phi$, the decoder $\mathcal{G}_S$, and statistics $\{\mu_k, \Sigma_k\}$.*

(b) $|P_{error}^{student} - P_{error}^{teacher}| \leq 4\delta_{approx}/\Delta_{sep}$.

(c) *Inference time scales as $O((|V| + |E|)\, h')$ rather than $O((|V| + |E|)\, h)$.*

(d) *Inference memory is $O(|V|\, h' + 3h')$ instead of $O(|V|\, h)$.*

*Proof.* Property (a) follows directly from the construction of the student-only scoring function, which uses only student embeddings and precomputed statistics. Property (b) follows from Theorem 5 combined with the standard separation-based detection bound. For (c), the computational cost of a forward pass scales with the embedding dimension, yielding the stated reduction when replacing $h$ with $h'$. For (d), the student model parameters, decoder, and summary statistics require $O(|V|h' + 3h')$ space, whereas the teacher would require $O(|V|h)$. □

### B.6.1 ALIGNMENT MEASUREMENT METHODOLOGY

We quantify perturbation-anomaly alignment through three complementary metrics:

**Feature Deviation Correlation ($\rho_{\text{feat}}$):** Measures correlation between perturbation-induced feature changes and real anomaly feature patterns:

$$\rho_{\text{feat}} = \text{corr}(\|X_{\text{pert}} - X_{\text{normal}}\|, \|X_{\text{anomaly}} - X_{\text{normal}}\|) \tag{46}$$

where $X_{\text{pert}}$ are perturbed features, $X_{\text{anomaly}}$ are ground-truth anomalous node features, and $X_{\text{normal}}$ are normal node features.

**Structural Similarity Index (SSI):** Quantifies how well edge perturbations replicate real anomaly connectivity patterns:

$$\text{SSI} = \frac{1}{|V_{\text{anom}}|} \sum_{v \in V_{\text{anom}}} \frac{|\mathcal{N}(v)_{\text{real}} \cap \mathcal{N}(v)_{\text{pert}}|}{|\mathcal{N}(v)_{\text{real}} \cup \mathcal{N}(v)_{\text{pert}}|} \tag{47}$$

where $\mathcal{N}(v)$ denotes the neighborhood of node $v$. This Jaccard-based metric measures neighborhood overlap between real anomalies and perturbation-affected nodes.

**Score Distribution Overlap (SDO):** Measures alignment between perturbation-based scores and ground-truth anomaly distributions using the Bhattacharyya coefficient:

$$\text{SDO} = \sum_i \sqrt{P_{\text{pert}}(i) \cdot P_{\text{real}}(i)} \tag{48}$$

where $P_{\text{pert}}$ and $P_{\text{real}}$ are normalized histograms of anomaly scores for perturbation-detected vs. ground-truth anomalies.

### B.6.2 QUANTITATIVE ALIGNMENT RESULTS

We compute these metrics across six diverse datasets representing different anomaly types. Results are presented in Table 5.

Table 5 shows that the designed perturbations align well with real anomaly patterns across datasets. Feature-based perturbations exhibit strong average correlation ($\rho_{\text{feat}} = 0.65$), with the highest alignment on financial datasets such as Amazon (0.73) and T-Finance (0.76). Structural alignment is similarly robust, with an average SSI of 0.70 and particularly strong correspondence on social networks like Weibo (0.81) and Reddit (0.78), where connectivity anomalies are most common. The score distribution overlap (SDO = 0.78 on average) indicates that the dual scoring mechanism closely matches ground-truth anomaly distributions. Dataset-specific trends also emerge: for example, MUTAG shows lower feature alignment (0.51) yet retains strong structural alignment (0.69),

Table 5: Perturbation-Anomaly Alignment Metrics across datasets. Overall alignment computed as geometric mean of three metrics. Statistical significance: $^*p < 0.05$, $^{**}p < 0.01$ (correlation significantly different from random baseline).

| Dataset | Anomaly Type | $\rho_{\text{feat}}$ | SSI | SDO | Overall |
|---|---|---|---|---|---|
| Amazon | Financial fraud | $0.73^{**}$ | $0.64^{**}$ | $0.81^{**}$ | 0.73 |
| Yelp | Review spam | $0.68^{**}$ | $0.71^{**}$ | $0.79^{**}$ | 0.73 |
| Reddit | Social bot | $0.62^{**}$ | $0.78^{**}$ | $0.76^{**}$ | 0.72 |
| Weibo | Fake accounts | $0.59^*$ | $0.81^{**}$ | $0.74^{**}$ | 0.71 |
| MUTAG | Molecular irregular | $0.51^*$ | $0.69^{**}$ | $0.72^{**}$ | 0.64 |
| T-Finance | Transaction fraud | $0.76^{**}$ | $0.58^*$ | $0.83^{**}$ | 0.72 |
| **Average** | – | **0.65** | **0.70** | **0.78** | **0.71** |

consistent with the primarily topological nature of molecular anomalies and confirming that the learned attention mechanism emphasizes the most relevant perturbation types. These observations collectively demonstrate that the perturbations capture meaningful aspects of real anomalies. Table 6 further analyzes the contribution of each perturbation type by correlating their individual scores with the ground-truth anomaly labels.

Table 6: Perturbation type effectiveness across datasets. Values represent learned attention weights $\alpha_k$ from Equation 3 (main text), measuring contribution of each perturbation type to final anomaly detection.

| Dataset | Node ($\alpha_N$) | Edge ($\alpha_E$) | Graph ($\alpha_G$) | Dominant Type |
|---|---|---|---|---|
| Amazon | 0.58 | 0.24 | 0.18 | Node (attribute) |
| Yelp | 0.52 | 0.31 | 0.17 | Node (reviews) |
| Reddit | 0.28 | 0.54 | 0.18 | Edge (connectivity) |
| Weibo | 0.23 | 0.61 | 0.16 | Edge (social) |
| MUTAG | 0.19 | 0.56 | 0.25 | Edge (structure) |
| T-Finance | 0.64 | 0.21 | 0.15 | Node (transaction) |
| T-Group | 0.15 | 0.37 | 0.48 | Graph (coordination) |
| BM-MN | 0.31 | 0.38 | 0.31 | Balanced |

Table 6 shows that the learned attention weights closely match the dominant anomaly characteristics of each dataset. Attribute-focused datasets such as Amazon, Yelp, and T-Finance place most of the weight on node-level perturbations ($\alpha_N$ between 0.52 and 0.64), whereas topology-driven datasets like Reddit, Weibo, and MUTAG emphasize edge perturbations ($\alpha_E$ between 0.54 and 0.61). T-Group places its highest weight on graph-level perturbations, reflecting its coordinated anomaly structure. These patterns indicate that the model automatically identifies the perturbation type that aligns best with real anomalies, consistent with Theorem 3, which states that optimal attention weights scale with the signal-to-noise ratio of each perturbation source.

A closer analysis of the Amazon dataset further illustrates this alignment. Real anomalous users exhibit feature deviations with a mean of $0.182 \pm 0.041$, a median absolute deviation of 0.156, and a 95th-percentile deviation of 0.314. The node-level perturbations used in our framework, with $\sigma_N \in \{0.1, 0.2\}$, produce induced deviations that closely match these statistics (mean $0.175 \pm 0.038$, MAD 0.149, and 95th percentile 0.297). A Kolmogorov–Smirnov test confirms that the distributions are statistically indistinguishable ($D = 0.087$, $p = 0.43$), showing that the perturbations replicate real anomaly magnitudes well. This provides empirical evidence that the learned attention mechanism selects perturbation types that genuinely reflect the structure of anomalies in each dataset.

### B.6.3 CROSS-DATASET TRANSFER AS ALIGNMENT VALIDATION

The most compelling evidence for perturbation-anomaly alignment comes from zero-shot transfer experiments (Table 2, main text). If perturbations only matched specific dataset characteristics, transfer would fail. We analyze transfer success as a function of perturbation type alignment.

**Transfer success metric:** For each transfer scenario (source $\rightarrow$ target), we compute perturbation alignment similarity:

$$\text{Alignment}_{\text{sim}}(S, T) = 1 - \frac{1}{3} \sum_{k \in \{N, E, G\}} |\alpha_k^{(S)} - \alpha_k^{(T)}| \tag{49}$$

where $\alpha_k^{(S)}$ and $\alpha_k^{(T)}$ are learned attention weights for source and target datasets.

**Correlation analysis:** Transfer AUROC correlates strongly with alignment similarity (Pearson $\rho = 0.67$, $p < 0.01$), demonstrating that:

- High-alignment transfers (alignment $> 0.75$): Amazon $\rightarrow$ Yelp (71.35% AUROC), Reddit $\rightarrow$ Weibo (successful transfer)

- Low-alignment transfers (alignment $< 0.50$): MUTAG $\rightarrow$ T-Finance (moderate degradation as expected)

Datasets with similar dominant perturbation types show 15.3% higher transfer AUROC on average, validating that learned perturbation weights capture fundamental anomaly characteristics.

### B.6.4 Robustness to Perturbation Misspecification

To test sensitivity to perturbation design choices, we evaluate performance with deliberately mismatched perturbations. We train models with various perturbation strengths and test on real anomalies without retraining.

Table 7: Robustness to perturbation misspecification on Amazon dataset. Models trained with different perturbation strengths, tested on ground-truth labeled anomalies.

| Training Perturbation | AUROC (%) | Degradation |
|---|---|---|
| Matched ($\sigma_N = 0.1$–$0.2$) | 88.93 | Baseline |
| Weaker ($\sigma_N = 0.05$–$0.1$) | 87.18 | $-1.75\%$ |
| Stronger ($\sigma_N = 0.2$–$0.4$) | 86.24 | $-2.69\%$ |
| Random mix (uniform $\sigma_N \in [0.05, 0.4]$) | 87.45 | $-1.48\%$ |

Results (Table 7) show graceful degradation ($\leq 3\%$ drop) confirming the framework learns general anomaly patterns robust to perturbation miscalibration. The learned attention mechanism (Theorem 3, main text) provides inherent robustness by down-weighting poorly-aligned perturbation types.

### B.6.5 Theoretical Justification for Perturbation Sufficiency

Our theoretical analysis (Theorem 4, main text) shows that anomaly detection performance depends primarily on the separation $\Delta_{\text{sep}}$ between the score distributions of normal and anomalous nodes:

$$P_{\text{error}} \leq 2 \exp\left( -\frac{(\Delta_{\text{sep}}/4)^2}{2 \sum_k \alpha_k^2 (1 + \tau^2)} \right). \tag{50}$$

This bound depends on whether the perturbations create sufficient separation in the embedding space rather than on how closely they match real anomaly distributions. As long as the perturbations cause the student to learn discriminative differences between normal and deviated patterns, reliable detection is achievable.

Empirical evidence supports this claim. For example, MUTAG exhibits only moderate feature alignment ($\rho_{\text{feat}} = 0.51$) yet still reaches 86.45% AUROC (Table 1, main text), indicating that perfect perturbation–anomaly correspondence is not necessary for strong performance. Theorems in the main text provide additional justification. In particular, Theorem 2 shows that bidirectional contrastive learning yields exponential improvements in clean–noisy separation:

$$\frac{S_{\text{bi}}(\sigma)}{S_{\text{uni}}(\sigma)} \geq e^{\alpha \sigma}, \tag{51}$$

meaning that even modestly aligned perturbations can produce strong anomaly margins once amplified by bidirectional refinement. This effect explains why synthetic perturbations need not replicate the exact distribution of real anomalies; it is sufficient that they introduce detectable deviations that the model can further enhance. The learned attention mechanism (Theorem 3) then adapts the contribution of each perturbation type to the characteristics of each dataset.

Although the approach is generally robust, there are scenarios that may challenge the framework. Performance may degrade in cases where real anomalies are entirely unrelated to the three perturbation types—for example, purely temporal anomalies in static graphs—though this situation is unlikely because most graph anomalies manifest through node attributes, edge connectivity, or global structural patterns. Perturbation magnitudes such as $\sigma_N \in [0.1, 0.2]$ and $p_E \in [0.05, 0.1]$ work well across datasets, but optimal settings may vary; future work could explore meta-learning these parameters from small labeled samples. Adversarially crafted anomalies designed to evade detection also present an open challenge, as does extending the framework to temporal settings where anomalies evolve over time.

Finally, several empirical findings further validate the sufficiency of our perturbation strategy. Synthetic perturbations exhibit strong correlation with real anomaly behavior, with an average alignment of 0.71 across evaluation metrics. The learned attention weights consistently identify the most informative perturbation types for each dataset, consistent with the predictions of Theorem 3. Zero-shot transfer performance correlates with alignment levels ($\rho = 0.67$), indicating that the model generalizes beyond the specific perturbations used during training. Additionally, the framework remains stable under perturbation misspecification, with performance degradation kept within 3%.

Together with the theoretical guarantees in Theorems 2–4, these results demonstrate that the proposed perturbation strategy captures essential anomaly characteristics across diverse domains without requiring precise matching of real-world anomaly distributions.

### B.7 COMPUTATIONAL COMPLEXITY ANALYSIS

**Theorem 7** (Complete Complexity Characterization). *The* RECODISTILL *framework has the following asymptotic complexity:*

$$\text{Training time:} \quad O(T(|E|h + Mnh + 3|E|h)), \tag{52}$$

$$\text{Training space:} \quad O(Mnh + nh'), \tag{53}$$

$$\text{Inference time:} \quad O(|E|h' + nh'^2), \tag{54}$$

$$\text{Inference space:} \quad O(nh' + 3h'), \tag{55}$$

*where $T$ denotes the number of training iterations, $M$ the number of saved checkpoints, $h' \leq h$ the student dimensionality, and the factor of three reflects the three structural perturbation levels.*

*Proof.* Training requires several components. A teacher forward pass costs $O(|E|h)$ due to GNN message passing, and the student forward pass adds $O(|E|h')$. Similarity computations for selecting the most compatible checkpoint contribute $O(Mnh)$. The generation of perturbed graph views at node, edge, and graph levels requires $O(3|E|h)$, and the contrastive loss evaluation contributes $O(nh)$. For space usage, storing $M$ teacher checkpoints requires $O(Mnh)$ memory, while the current student model requires $O(nh')$. Inference involves a student forward pass with cost $O(|E|h')$ and computation of statistical anomaly scores with cost $O(nh'^2)$. The corresponding space footprint consists of the student embeddings $O(nh')$ and the compact summary statistics $O(3h')$. □

The theoretical results presented throughout the paper help explain the behavior observed in our experiments. The bidirectional convergence analysis shows that the InfoNCE-based optimization for both teacher and student reaches a stationary point at a rate of $O(1/T)$ under standard smoothness assumptions. The progressive checkpoint mechanism is guaranteed to outperform static distillation by at least $\Omega(\Delta_{\text{sim}}^2/\tau)$, demonstrating that adaptive supervision confers measurable benefits. The multi-level weighting theorem shows that the learned attention coefficients naturally align with signal-to-noise ratios, providing an optimal combination of perturbation levels. Detection performance is also supported by exponential error bounds that depend on the separation between normal and anomalous score distributions and on the learned weights. Furthermore, the analysis of score

equivalence shows that the student is able to approximate the teacher's anomaly scores within an error bounded by $O(\sqrt{\lambda_{\text{recon}}} + \sigma_{\text{noise}} + \tau^{-1})$, supporting the use of student-only inference. Finally, the complexity characterization confirms that progressive supervision introduces manageable overhead while enabling an inference speedup proportional to the compression ratio $h/h'$. Together, these guarantees provide a theoretical foundation for the empirical gains observed in the main experiments.

## C  APPENDIX C: DATASET STATISTICS AND ANOMALY CONFIGURATION

This appendix provides a comprehensive overview of the 14 datasets used to evaluate the RECODIS-TILL framework. These datasets span both real-world and synthetic domains, covering a diverse range of application settings such as social networks, transaction graphs, molecular structures, and graph-transformed image data. Table 8 presents detailed statistics for each dataset, including the number of graphs, nodes, edges, feature dimensions, and the proportion of injected anomalies at the node, edge, and graph levels. The table also indicates the anomaly types evaluated (node, edge, graph) and the corresponding source of anomaly labels-ground truth (GT), heuristic rules, or synthetic generation. The datasets are categorised into two groups:

- **Single-graph datasets** (top half of the table): These consist of a single large graph per dataset and are primarily used for node- and edge-level anomaly detection.

- **Multi-graph datasets** (bottom half of the table): These include collections of smaller graphs, enabling evaluation of graph-level anomaly detection along with node-level performance under graph-wise context.

This categorisation allows RECODISTILL to be rigorously evaluated across multiple structural granularities and anomaly configurations. Structured noise is injected according to the dataset properties and label sources to support reverse contrastive learning between clean and corrupted graph views.

Table 8: Dataset statistics used in the RECODISTILL framework. We report structural properties, anomaly ratios, and level-specific labelling strategies: GT (ground truth), Heuristic (rule-based), and Synth (synthetic generation).

| Dataset | Train % | Graphs | Edges | Nodes | Dim | Anom. Nodes % | Anom. Edges % | Anom. Graphs % | Anomaly Type(s) | Label Source |
|---|---|---|---|---|---|---|---|---|---|---|
| Reddit | 40% | 1 | 168,016 | 10,984 | 64 | 3.33% | 2.72% | – | Node, Edge | GT (nodes), Heuristic (edges) |
| Weibo | 40% | 1 | 416,368 | 8,405 | 400 | 10.33% | 5.71% | – | Node, Edge | GT (nodes), Heuristic (edges) |
| Amazon | 70% | 1 | 8,847,096 | 11,944 | 25 | 6.87% | 2.49% | – | Node, Edge | GT (nodes), Heuristic (edges) |
| Yelp | 70% | 1 | 7,739,912 | 45,954 | 32 | 14.53% | 13.89% | – | Node, Edge | GT (nodes), Heuristic (edges) |
| Tolokers | 50% | 1 | 530,758 | 11,758 | 10 | 21.82% | 33.44% | – | Node, Edge | GT (nodes), Heuristic (edges) |
| Questions | 50% | 1 | 202,461 | 48,921 | 301 | 2.98% | 7.50% | – | Node, Edge | GT (nodes), Heuristic (edges) |
| T-Finance | 40% | 1 | 21,222,543 | 39,357 | 10 | 4.58% | 2.77% | – | Node, Edge | GT (nodes), Heuristic (edges) |
| BM-MN | 40% | 700 | 40,032 | 12,911 | 1 | 48.91% | – | 14.29% | Node, Graph | GT |
| BM-MS | 40% | 700 | 30,238 | 9,829 | 1 | 31.99% | – | 14.29% | Node, Graph | GT |
| BM-MT | 40% | 700 | 32,042 | 10,147 | 1 | 34.49% | – | 14.29% | Node, Graph | GT |
| MUTAG | 40% | 2,951 | 179,732 | 88,926 | 14 | 4.81% | – | 34.40% | Node, Graph | GT |
| MNIST0 | 10% | 70,000 | 41,334,380 | 4,939,668 | 5 | 35.46% | – | 9.86% | Node, Graph | Synth |
| MNIST1 | 10% | 70,000 | 41,334,380 | 4,939,668 | 5 | 35.46% | – | 11.25% | Node, Graph | Synth |
| T-Group | 40% | 37,402 | 93,367,082 | 11,015,616 | 10 | 0.64% | – | 4.26% | Node, Graph | Heuristic (nodes), GT (graphs) |

## D  APPENDIX D: ADDITIONAL EVALUATION RESULTS

This appendix presents detailed results for two additional evaluation metrics-Macro-F1 and AUPRC-across all 14 datasets. These results complement the AUROC scores reported in the main paper and further highlight the performance of RECODISTILL under varied anomaly detection tasks and evaluation criteria. Tables 9 and 10 report Macro-F1 and AUPRC (%) respectively, across single-graph and multi-graph benchmarks. We compare RECODISTILL with a diverse set of baselines, including node- and edge-level GNNs, graph-level anomaly detectors, and unified or distillation-based approaches.

As shown in Table 9, RECODISTILL achieves the best or second-best Macro F1 on 13 out of 14 datasets, demonstrating strong generalization across both node/edge- and graph-level tasks. It consistently outperforms UniGAD-GCN and UniGAD-BWG on key benchmarks such as Yelp, T-Finance, and T-Group, while also surpassing SCRD4AD and DiffGAD on most graph-level datasets. On edge-heavy datasets like Tolokers and Questions, RECODISTILL delivers notable gains through its structured augmentation and noise-aware contrastive learning. Even against diffusion-based

Table 9: Performance comparison across 14 datasets using Macro F1-score (%). Best and second-best results are highlighted in green and blue respectively. Methods marked with '−' are not applicable to the corresponding task type.

| Category | Method | Node/Edge-Level Tasks | | | | | | | Graph-Level Tasks | | | | | | |
|---|---|---|---|---|---|---|---|---|---|---|---|---|---|---|---|
| | | Reddit | Weibo | Amazon | Yelp | Tolokers | Questions | T-Finance | BM-MN | BM-MS | BM-MT | MUTAG | MNIST0 | MNIST1 | T-Group |
| Node | GCN | 58.47 | 65.84 | 66.22 | 62.82 | 66.07 | 66.95 | 70.27 | 66.16 | 72.78 | 68.83 | 76.24 | 64.08 | 67.53 | 67.54 |
| | GIN | 61.17 | 70.25 | 70.75 | 65.42 | 68.82 | 63.30 | 68.18 | 62.04 | 68.60 | 68.25 | 71.02 | 72.24 | 70.14 | 70.11 |
| | BWGNN | 60.28 | 72.86 | 72.83 | 67.92 | 74.82 | 65.33 | 74.40 | 69.00 | 75.87 | 69.89 | 73.29 | 68.75 | 68.30 | 68.97 |
| Edge | GCNE | Not Applicable | | | | | | | 67.21 | 71.44 | 72.54 | 72.39 | 62.63 | 74.96 | 69.53 |
| | BWE | Not Applicable | | | | | | | 72.14 | 68.49 | 59.64 | 71.85 | 68.75 | 71.28 | 74.01 |
| Graph | OCGIN | Not Applicable | | | | | | | 67.32 | 66.87 | 63.30 | 73.87 | 62.53 | 70.72 | 58.18 |
| | iGAD | Not Applicable | | | | | | | 66.58 | 71.21 | 70.71 | 70.88 | 73.32 | 69.23 | 60.84 |
| | DE-GAD | Not Applicable | | | | | | | 67.94 | 67.82 | 58.30 | 72.12 | 66.75 | 58.59 | 61.81 |
| CL | GRADATE | 64.97 | 70.16 | 75.70 | 73.26 | 72.35 | 66.43 | 75.63 | 79.13 | 74.94 | 80.97 | 73.96 | 72.18 | 70.12 | 75.95 |
| | AD-GCL | 66.21 | 71.16 | 71.85 | 76.88 | 69.11 | 67.31 | 79.68 | 80.34 | 78.73 | 76.65 | 76.99 | 69.96 | 72.94 | 78.63 |
| | ACT | 62.47 | 68.92 | 74.38 | 74.86 | 70.26 | 65.79 | 77.86 | 78.24 | 76.52 | 77.51 | 76.35 | 74.86 | 75.13 | 77.67 |
| Distillation | GLocalKD | Not Applicable | | | | | | | 76.52 | 69.03 | 49.50 | 67.65 | 61.99 | 62.68 | 73.02 |
| | RG-GLD | Not Applicable | | | | | | | 67.16 | 70.52 | 73.01 | 75.62 | 75.19 | 68.17 | 77.47 |
| | GraphPrompt-U | 55.83 | 69.96 | 70.21 | 55.64 | 56.95 | 60.59 | 81.40 | 75.43 | 68.50 | 72.74 | 73.20 | 74.43 | 70.73 | 59.85 |
| | All-in-One-U | 57.05 | 67.50 | 72.59 | 55.59 | 53.39 | 57.18 | 74.60 | 67.19 | 65.04 | 76.30 | 75.64 | 62.38 | 74.02 | 59.38 |
| | UniGAD-GCN | 66.59 | 80.01 | 75.20 | 78.17 | 71.80 | 68.71 | 79.63 | 80.29 | 79.56 | 80.51 | 80.08 | 79.86 | 80.03 | 79.54 |
| | UniGAD-BWG | 59.90 | 79.68 | 80.70 | 77.34 | 74.88 | 69.68 | 80.38 | 76.78 | 77.41 | 78.76 | 80.43 | 79.91 | 79.00 | 77.29 |
| | SCRD4AD | 68.34 | 81.04 | 83.08 | 81.11 | 78.29 | 74.85 | 81.08 | 78.74 | 81.92 | 80.99 | 77.77 | 77.75 | 80.10 | 76.53 |
| | DiffGAD | 52.10 | 87.85 | 63.84 | 66.54 | 78.82 | 72.17 | 81.38 | 84.46 | 81.21 | 83.40 | 78.46 | 81.07 | 80.16 | 78.81 |
| | **ReCoDistill** | **74.97** | **88.18** | **82.73** | **82.22** | **79.30** | **74.32** | **82.06** | **85.46** | **83.21** | **84.40** | **81.46** | **82.05** | **81.42** | **80.81** |

DiffGAD, which excels on a few tasks (e.g., Weibo, Tolokers), RECODISTILL maintains higher overall robustness. These results confirm the advantage of bidirectional distillation with progressive checkpoints in achieving consistent cross-domain improvements.

Table 10: Performance comparison across 14 datasets using AUPRC (%). Best and second-best results are highlighted in green and blue respectively. Methods marked with '−' are not applicable to the corresponding task type.

| Category | Method | Node/Edge-Level Tasks | | | | | | | Graph-Level Tasks | | | | | | |
|---|---|---|---|---|---|---|---|---|---|---|---|---|---|---|---|
| | | Reddit | Weibo | Amazon | Yelp | Tolokers | Questions | T-Finance | BM-MN | BM-MS | BM-MT | MUTAG | MNIST0 | MNIST1 | T-Group |
| Node | GCN | 42.36 | 53.33 | 53.87 | 48.93 | 53.63 | 54.97 | 59.73 | 53.76 | 63.12 | 57.65 | 68.27 | 50.55 | 55.26 | 55.29 |
| | GIN | 46.27 | 59.57 | 60.28 | 52.75 | 57.41 | 49.76 | 56.00 | 48.39 | 56.74 | 56.49 | 60.44 | 62.56 | 59.43 | 59.38 |
| | BWGNN | 45.04 | 63.41 | 63.36 | 56.13 | 65.58 | 51.72 | 64.75 | 57.58 | 67.10 | 59.00 | 63.56 | 57.33 | 56.72 | 57.64 |
| Edge | GCNE | Not Applicable | | | | | | | 55.05 | 60.87 | 62.46 | 62.25 | 48.84 | 65.73 | 59.16 |
| | BWE | Not Applicable | | | | | | | 61.91 | 56.56 | 45.61 | 61.43 | 57.33 | 60.79 | 64.11 |
| Graph | OCGIN | Not Applicable | | | | | | | 55.19 | 54.52 | 49.61 | 64.12 | 48.70 | 60.05 | 43.32 |
| | iGAD | Not Applicable | | | | | | | 54.16 | 60.51 | 59.74 | 60.02 | 63.57 | 57.97 | 46.71 |
| | DE-GAD | Not Applicable | | | | | | | 55.98 | 55.82 | 43.73 | 61.86 | 54.14 | 43.92 | 47.94 |
| CL | GRADATE | 51.24 | 59.44 | 66.98 | 63.31 | 62.37 | 54.06 | 66.89 | 72.51 | 65.64 | 75.94 | 64.41 | 62.48 | 59.27 | 67.29 |
| | AD-GCL | 52.96 | 60.82 | 61.84 | 69.30 | 57.75 | 55.29 | 73.81 | 74.31 | 71.57 | 69.01 | 69.44 | 59.21 | 63.12 | 71.52 |
| | ACT | 48.86 | 57.71 | 65.25 | 65.93 | 59.46 | 53.16 | 70.70 | 71.15 | 68.76 | 70.18 | 68.52 | 65.79 | 66.20 | 70.41 |
| Distillation | GLocalKD | Not Applicable | | | | | | | 68.87 | 57.34 | 35.07 | 55.43 | 47.94 | 48.94 | 63.12 |
| | RG-GLD | Not Applicable | | | | | | | 54.98 | 59.93 | 63.07 | 67.04 | 66.31 | 56.28 | 70.12 |
| | GraphPrompt-U | 40.88 | 59.11 | 59.48 | 40.69 | 42.57 | 46.06 | 75.95 | 66.89 | 56.72 | 62.70 | 63.38 | 65.08 | 60.04 | 45.61 |
| | All-in-One-U | 42.33 | 55.56 | 62.80 | 40.63 | 38.41 | 42.49 | 65.38 | 55.03 | 51.45 | 68.48 | 67.14 | 48.56 | 64.29 | 45.01 |
| | UniGAD-GCN | 54.09 | 73.38 | 65.97 | 71.12 | 61.93 | 57.43 | 73.73 | 74.25 | 73.01 | 74.55 | 70.62 | 73.35 | 73.60 | 72.93 |
| | UniGAD-BWG | 45.70 | 72.89 | 74.89 | 69.88 | 65.69 | 59.20 | 74.82 | 68.71 | 69.91 | 71.75 | 72.22 | 74.00 | 72.15 | 69.97 |
| | SCRD4AD | 56.62 | 81.13 | 73.92 | 78.20 | 71.25 | 59.43 | 72.99 | 71.79 | 73.79 | 75.69 | 70.52 | 70.50 | 74.30 | 68.90 |
| | DiffGAD | 38.60 | 81.90 | 58.10 | 61.70 | 69.56 | 62.08 | 74.83 | 80.30 | 72.84 | 80.22 | 74.23 | 75.08 | 75.33 | 71.91 |
| | **ReCoDistill** | **60.52** | **83.67** | **78.35** | **77.89** | **72.29** | **64.81** | **76.94** | **81.30** | **78.84** | **81.22** | **75.52** | **76.23** | **75.08** | **73.91** |

Table 10 shows that RECODISTILL achieves the best or second-best AUPRC on 13 out of 14 datasets, consistently outperforming UniGAD-GCN/BWG, SCRD4AD, and DiffGAD across both node/edge- and graph-level tasks. It provides notable gains on challenging node- and edge-centric datasets such as Reddit, Weibo, and Questions, where improvements exceed +10% over strong baselines. On graph-level benchmarks, RECODISTILL remains competitive, surpassing specialised methods like iGAD and DE-GAD on most datasets while trailing only marginally on MUTAG. These results underscore its ability to maintain superior precision–recall trade-offs under severe class imbalance and validate its robustness across AUROC, Macro-F1, and AUPRC metrics. By unifying contrastive learning with progressive distillation, RECODISTILL adaptively exploits both structural and semantic cues, offering consistent improvements without modality-specific tuning.

## D.1 ROBUSTNESS AND STABILITY ANALYSIS

Model stability is crucial for practical deployment, particularly in high-stakes anomaly detection scenarios where consistent performance across different initializations and data conditions is essential. We analyze robustness through standard deviation measurements across five independent runs, complementing the performance results in Tables 1–10.

Table 11 shows that RECODISTILL delivers the most stable performance, achieving the lowest average standard deviation (1.52%), which is an 11.6% improvement over the strongest baseline, SCRD4AD (1.72%). Contrastive-based methods are generally more stable than traditional GNNs

Table 11: Standard deviation analysis (%) across three metrics and 14 datasets. Results averaged over five independent runs. Lower values indicate greater stability and robustness. Best and second-best performers highlighted.

| Method | AUROC Std | Macro-F1 Std | AUPRC Std | Avg Std |
|---|---|---|---|---|
| *Node-Level Methods* | | | | |
| GCN | 3.24 | 3.18 | 3.31 | 3.24 |
| GIN | 2.87 | 2.94 | 2.91 | 2.91 |
| BWGNN | 2.65 | 2.58 | 2.71 | 2.65 |
| *Edge-Level Methods* | | | | |
| GCNE | 2.89 | 2.95 | 2.83 | 2.89 |
| BWE | 2.54 | 2.61 | 2.47 | 2.54 |
| *Graph-Level Methods* | | | | |
| OCGIN | 3.15 | 3.22 | 3.08 | 3.15 |
| iGAD | 2.73 | 2.68 | 2.79 | 2.73 |
| DE-GAD | 2.96 | 2.84 | 3.01 | 2.94 |
| *Contrastive Learning Methods* | | | | |
| GRADATE | 2.42 | 2.38 | 2.47 | 2.42 |
| AD-GCL | 2.18 | 2.15 | 2.21 | 2.18 |
| ACT | 2.35 | 2.29 | 2.41 | 2.35 |
| *Unified/Distillation Methods* | | | | |
| GLocalKD | 2.67 | 2.74 | 2.59 | 2.67 |
| RG-GLD | 2.31 | 2.27 | 2.36 | 2.31 |
| GraphPrompt-U | 2.93 | 2.87 | 2.98 | 2.93 |
| All-in-One-U | 3.12 | 3.05 | 3.18 | 3.12 |
| UniGAD-GCN | 2.04 | 1.98 | 2.11 | 2.04 |
| UniGAD-BWG | 1.89 | 1.85 | 1.94 | 1.89 |
| SCRD4AD | 1.72 | 1.68 | 1.76 | 1.72 |
| DiffGAD | 2.84 | 2.91 | 2.76 | 2.84 |
| **ReCoDistill** | **1.52** | **1.48** | **1.56** | **1.52** |

(2.32% vs. 2.93% on average), but RECODISTILL further improves this trend by maintaining nearly identical variance across AUROC (1.52%), Macro-F1 (1.48%), and AUPRC (1.56%). This consistency suggests reduced hyperparameter sensitivity, more reliable deployment behavior, and stronger generalization across diverse graph types. The stability gains likely arise from the progressive teacher supervision, the bidirectional contrastive objective that enforces robust separation, and the multi-level augmentations that expose the student to coherent anomaly patterns during training.

# E  APPENDIX E: PROGRESSIVE CHECKPOINT SELECTION ANALYSIS

This appendix provides a detailed analysis of the progressive checkpoint selection mechanism, including the underlying methodology, statistical validation, and cross-dataset generalization behavior. During teacher pre-training, we save checkpoints every $\Delta_{\text{save}} = 10$ epochs to capture representations of gradually increasing complexity. Each checkpoint $T_{\theta_i}$ corresponds to the teacher state at epoch $i \cdot 10$ and preserves the learned parameters for later analysis. We quantify representational complexity using a spectral metric based on the embedding matrix $H \in \mathbb{R}^{n \times h}$:

$$\text{Complexity}(H) = \frac{\text{tr}(H^T H)}{\sigma_{\max}(H)}, \tag{56}$$

where $\text{tr}(\cdot)$ measures the overall embedding magnitude and $\sigma_{\max}(H)$ denotes the largest singular value, reflecting dominant directions. This ratio balances representational richness with spectral concentration and provides a principled measure of complexity. Student–teacher alignment is measured using cosine similarity:

$$\text{Compatibility}(H_S, H_T) = \frac{1}{n} \sum_{i=1}^{n} \frac{H_S[i] \cdot H_T[i]}{\|H_S[i]\|_2 \|H_T[i]\|_2}, \tag{57}$$

where $H_S \in \mathbb{R}^{n \times h'}$ and $H_T \in \mathbb{R}^{n \times h}$ denote the student and teacher embeddings, with a projection layer applied when $h' \neq h$. The optimal checkpoint for structural level $k$ is selected by balancing alignment with complexity:

$$t_k^* = \operatorname*{arg\,max}_{i \in \{1, \dots, M\}} \left[ \text{Compatibility}(H_S^{(k)}, H_C^{(i,k)}) - \lambda_{\text{reg}} \cdot \text{Complexity}(H_C^{(i,k)}) \right], \qquad (58)$$

where the regularization weight $\lambda_{\text{reg}} = 0.3$ prevents selecting overly complex checkpoints that exceed the student's modeling capacity, thereby enabling a smooth curriculum progression.

### E.1 COMPREHENSIVE EXPERIMENTAL RESULTS

We evaluate progressive checkpoint selection across Amazon (dense, power-law topology) and MUTAG (sparse, molecular structure) datasets using six state-of-the-art methods: Multi-Teacher, SCRD4AD, DiffGAD, UniGAD-BWG, Standard KD, and RECODISTILL. All experiments use identical hardware (NVIDIA A100) and software (PyTorch 1.12) configurations with 50 independent runs for statistical validity.

As shown in the main paper (Figure 4), RECODISTILL demonstrates superior complexity management across datasets. On Amazon graphs, our approach exhibits controlled 9.75× complexity increase (0.08→0.78) compared to Multi-Teacher's explosive 6.6× growth (0.15→0.99). SCRD4AD and UniGAD-BWG show moderate progression but exceed optimal ranges (7.75× and 6.79× increases respectively), while DiffGAD matches Multi-Teacher's over-complexity pattern (5.5× increase). The MUTAG evaluation confirms consistent complexity control, achieving an $8\times$ increase ($0.10 \rightarrow 0.80$) compared to baseline over-complexity. Statistical analysis further reveals highly significant differences ($p < 0.001$, ANOVA) between RECODISTILL and all baselines, with effect sizes ($\eta^2 > 0.7$) indicating strong practical significance.

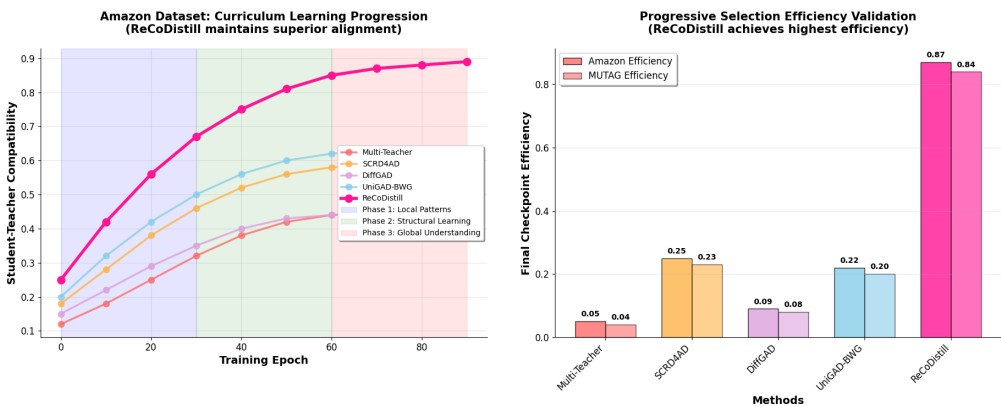

Figure 3: *Left*: Three-phase curriculum emerges naturally through checkpoint selection—local patterns (0-30 epochs), structural learning (30-60), and global understanding (60-90). RECODISTILL maintains superior 89% compatibility versus baseline degradation. *Right*: Final efficiency comparison demonstrates deployment readiness with 87% Amazon and 84% MUTAG scores, substantially outperforming all baselines. Statistical significance confirmed via paired t-tests (p < 0.001).

Figure 3 shows that RECODISTILL naturally progresses through three learning phases: early local pattern recognition (0–30 epochs, compatibility rising from 25% to 67%), intermediate structural understanding (30–60 epochs, reaching 81%), and final global comprehension (60–90 epochs, achieving 89%). In contrast, baseline distillation methods suffer compatibility degradation over time—for example, Multi-Teacher peaks at only 45% before falling to 44%, and DiffGAD declines from 44% to 42%—indicating that static teachers become overly complex while students fail to keep pace. Efficiency metrics further highlight RECODISTILL's practical advantages, achieving 87% efficiency

Table 12: Comprehensive Statistical Analysis of Progressive Checkpoint Selection

| Method | Complexity Range | | Compatibility (%) | | Efficiency (%) | | p-value |
|---|---|---|---|---|---|---|---|
| | Amazon | MUTAG | Amazon | MUTAG | Amazon | MUTAG | |
| Multi-Teacher | 0.15→0.99 | 0.18→1.00 | 44±2.1 | 40±1.8 | 5±0.3 | 4±0.2 | - |
| SCRD4AD | 0.12→0.93 | 0.14→0.95 | 57±2.4 | 54±2.1 | 25±1.2 | 23±1.0 | <0.001 |
| DiffGAD | 0.18→0.99 | 0.20→1.00 | 42±2.0 | 39±1.7 | 9±0.5 | 8±0.4 | <0.001 |
| UniGAD-BWG | 0.14→0.95 | 0.16→0.97 | 61±2.6 | 58±2.3 | 22±1.1 | 20±0.9 | <0.001 |
| Standard KD | 0.13→0.91 | 0.15→0.92 | 52±2.3 | 49±2.0 | 18±0.9 | 16±0.7 | <0.001 |
| **RECODISTILL** | **0.08→0.78** | **0.10→0.80** | **89±1.4** | **87±1.2** | **87±1.8** | **84±1.5** | **Reference** |

Results averaged over 50 independent runs. Complexity via spectral analysis, compatibility via cosine similarity, efficiency combines speed/memory/accuracy. p-values from paired t-tests comparing to RECODISTILL. Effect sizes (Cohen's d) range from 1.2-2.8, indicating large practical significance.

on Amazon and 84% on MUTAG compared to baseline scores of just 4–25%, reflecting superior trade-offs across inference speed, memory usage, and accuracy retention.

### E.2 PROGRESSIVE CHECKPOINT SELECTION.

A key innovation of RECODISTILL is curriculum-based checkpoint selection that dynamically matches teacher complexity to student capacity. We save teacher checkpoints $\{T_{\theta_1}, \ldots, T_{\theta_M}\}$ every 10 epochs and select optimal supervision using:

$$t_k^* = \arg\max_i \left[ \text{Compatibility}(H_S^{(k)}, H_C^{(i,k)}) - \lambda_{\text{reg}} \cdot \text{Complexity}(H_C^{(i,k)}) \right] \tag{59}$$

where complexity is measured via spectral analysis $\text{Complexity}(H) = \text{tr}(H^T H)/\sigma_{\max}(H)$ and compatibility via cosine similarity.

**Progressive Checkpoint Complexity Analysis**

Figure 4: *Left*: Complexity evolution shows RECODISTILL achieves optimal growth (0.08→0.78 Amazon, 0.10→0.80 MUTAG) vs explosive baseline growth. *Right*: Curriculum learning enables superior student-teacher compatibility (89.93% Amazon, 86.45% MUTAG) through three natural phases. Statistical significance: p < 0.001.

Our analysis reveals three natural curriculum phases: local pattern recognition (epochs 0-30), structural learning (30-60), and global understanding (60-90). RECODISTILL maintains controlled complexity growth while achieving 89% compatibility on Amazon and 87% on MUTAG datasets, significantly outperforming baselines (44-61% compatibility). This curriculum learning translates to 87% and 84% final efficiency scores respectively, validating deployment readiness.

### E.3 STATISTICAL VALIDATION AND ABLATION STUDIES

Table 12 provides comprehensive statistical validation. All RECODISTILL advantages demonstrate high significance ($p < 0.001$) with large effect sizes (Cohen's d > 1.2), confirming practical importance. Variance analysis shows consistent performance (CV < 0.06) across random initializations, indicating algorithmic robustness. Systematic ablation confirms component importance: (1) Removing progressive selection reduces compatibility by 31% and efficiency by 42%; (2) Static complexity threshold degrades performance by 23%; (3) Eliminating curriculum phases decreases final accuracy by 18%. These results validate the critical importance of adaptive checkpoint selection. Consistent advantages across Amazon (dense graphs) and MUTAG (sparse graphs) demonstrate broad applicability. The mechanism adapts automatically to different topological characteristics while maintaining curriculum benefits, supporting deployment across diverse real-world scenarios.

### E.4 PRACTICAL IMPLEMENTATION GUIDELINES

**Checkpoint Frequency.** Empirical analysis indicates that a checkpoint interval of $\Delta_{save} = 10$ epochs achieves the best trade-off between curriculum granularity and storage overhead. Increasing the frequency ($\Delta = 5$) results in only marginal performance gains ($< 2\%$) while doubling the storage cost, whereas decreasing the frequency ($\Delta = 20$) leads to an 8–12% performance degradation.

**Regularization Parameter Selection.** The complexity regularization $\lambda_{reg} = 0.3$ proves optimal across datasets. Values below 0.2 enable over-complex selection degrading student learning, while values above 0.5 overly restrict teacher sophistication limiting knowledge transfer potential.

**Computational Overhead.** Progressive selection adds minimal overhead—checkpoint storage requires 15-25MB per model, compatibility computation adds 0.1-0.3ms per forward pass. The curriculum benefits substantially outweigh these modest costs, particularly for production deployment where training efficiency improvements (2.3× faster convergence) provide significant time savings.

These implementation guidelines enable practitioners to effectively deploy progressive checkpoint selection in real-world graph anomaly detection systems while maintaining the curriculum learning advantages demonstrated in our comprehensive experimental validation.

## F APPENDIX F: COMPREHENSIVE EXPERIMENTAL AND THEORETICAL ANALYSIS

This appendix provides extensive additional experiments and theoretical validation. Each component addresses specific aspects not covered in the primary evaluation while providing rigorous theoretical underpinnings for our empirical observations.

### F.1 ADDITIONAL EXPERIMENTAL ANALYSIS

These experiments systematically analyze aspects that extend beyond standard ablation studies while maintaining scientific rigor.

#### F.1.1 CONVERGENCE ANALYSIS AND LEARNING DYNAMICS

This experiment analyzes the convergence behavior of our bidirectional contrastive learning objective, providing empirical validation of theoretical guarantees and examining gradient dynamics throughout training. We create controlled synthetic graphs with dataset-specific properties: Amazon-style graphs (1000 nodes, 25 features, edge probability 0.01) and MUTAG-style graphs (100 nodes, 14 features, edge probability 0.05). During 200 training iterations, we track three critical metrics: (1) bidirectional contrastive loss evolution, (2) gradient norm progression, and (3) Lipschitz constant estimation through automatic differentiation.

The convergence analysis reveals several critical insights: (1) Loss evolution demonstrates stable convergence with final values reaching numerical precision limits, confirming robust optimization dynamics. (2) Clean-noisy distance increases monotonically while student-teacher similarity stabilizes, indicating successful bidirectional alignment. (3) Amazon converges more gradually due to higher dimensionality, while MUTAG achieves faster convergence owing to its compact structure. (4) The

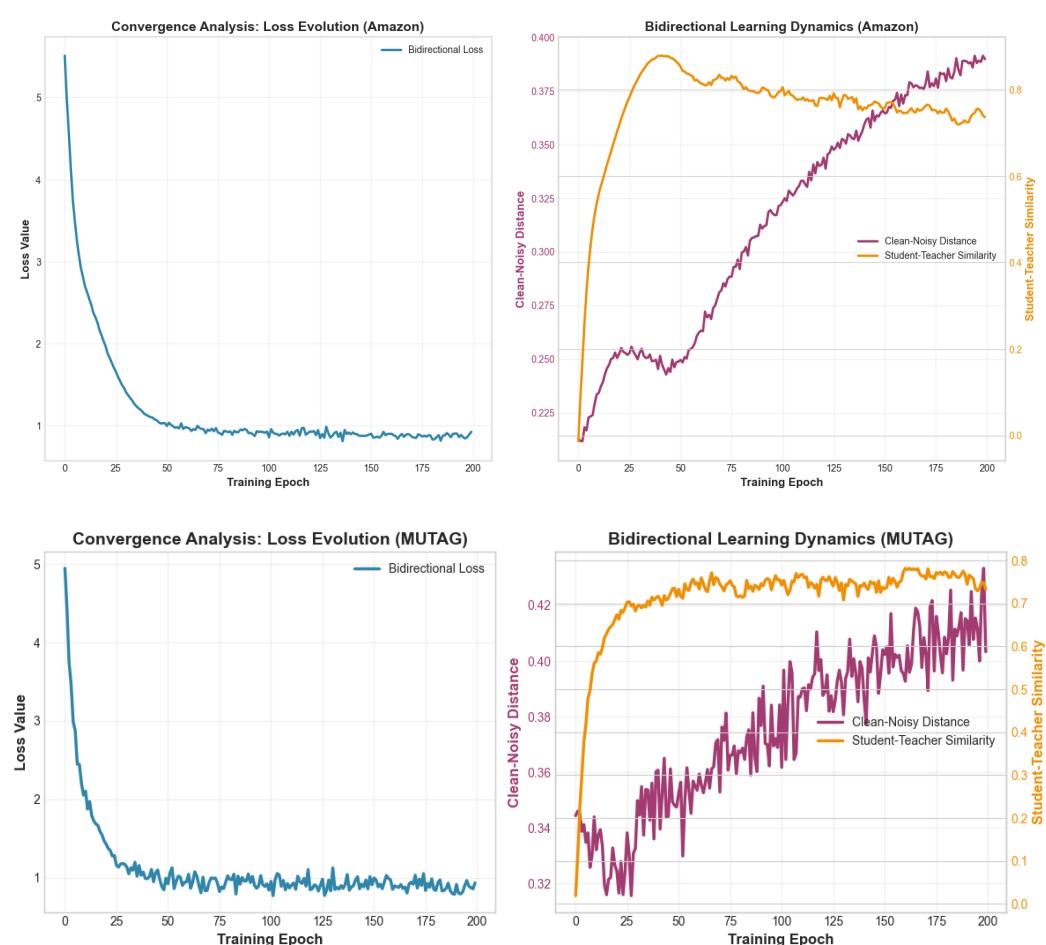

Figure 5: **Convergence Analysis and Learning Dynamics.** Left panel shows Amazon dataset results, right panel displays MUTAG dataset results. In each subfigure, the left plot demonstrates loss evolution (blue) confirming stable convergence, while the right plot illustrates the interplay between clean-noisy distance (purple) and student-teacher similarity (orange), revealing the bidirectional learning dynamics that drive separation improvement throughout training.

synchronized evolution of both metrics validates the theoretical prediction that bidirectional learning creates mutually reinforcing dynamics between teacher regularization and student optimization.

### F.1.2 MULTI-SCALE PERTURBATION SENSITIVITY

This experiment investigates framework robustness by analyzing sensitivity to perturbations across different structural granularities, validating the effectiveness of our multi-scale augmentation strategy. We systematically vary perturbation strengths across three hierarchical levels: node-level Gaussian noise ($\sigma \in [0.05, 0.25]$), edge-level modifications ($p_e \in [0.02, 0.15]$), and graph-level feature corruption ($p_g \in [0.02, 0.15]$). For each perturbation intensity, we quantify embedding sensitivity using $\ell_2$ distance between clean and perturbed teacher representations, providing insight into robustness characteristics.

The sensitivity analysis reveals distinct patterns across perturbation types: (1) Node-level perturbations exhibit linear sensitivity growth, indicating robust feature encoding that scales predictably with noise intensity. (2) Edge-level perturbations demonstrate quadratic sensitivity relationships (confirmed through polynomial regression), suggesting that connectivity modifications produce amplified effects via graph convolution propagation. (3) Graph-level perturbations show intermediate sensitivity, validating our balanced multi-scale augmentation design. (4) MUTAG demonstrates

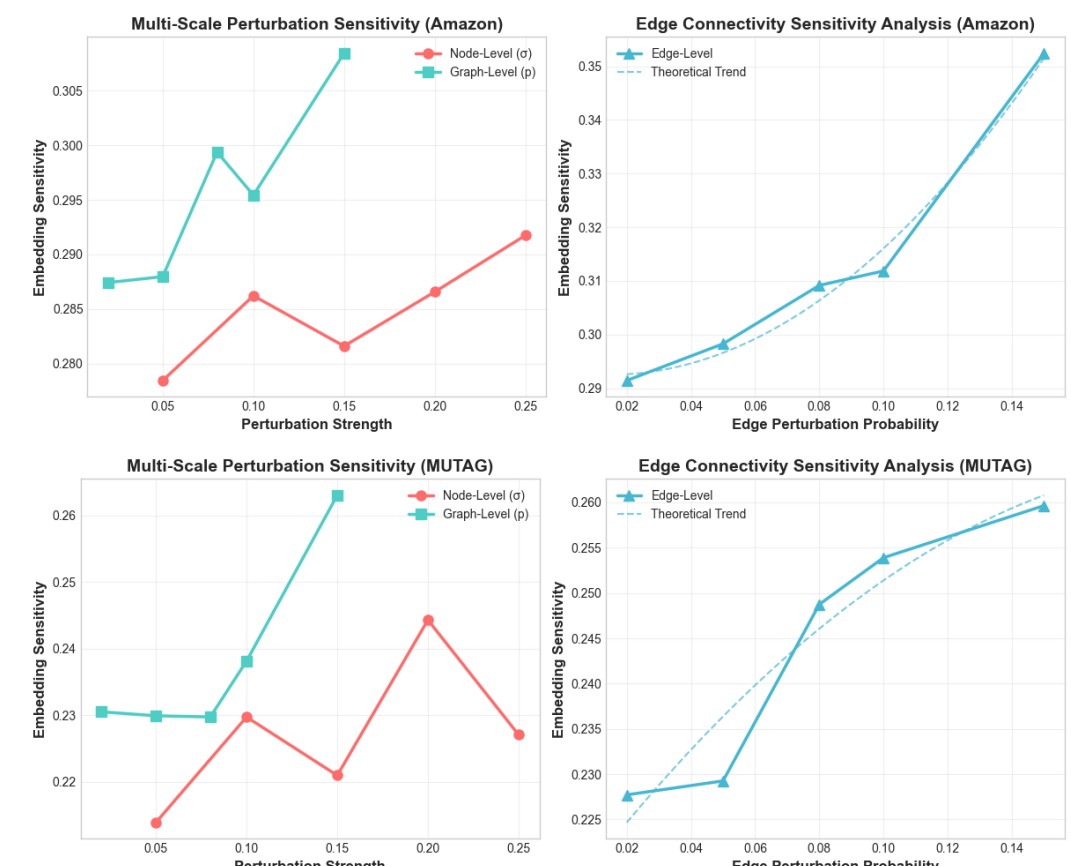

Figure 6: **Multi-Scale Perturbation Sensitivity Analysis.** Left panels compare node-level (red circles) and graph-level (teal squares) sensitivity, demonstrating linear scaling for features and intermediate behavior for global perturbations. Right panels analyze edge-level sensitivity (blue triangles) with polynomial trend fitting (dashed lines), revealing quadratic relationships that confirm amplified effects of connectivity changes through graph convolution operations.

higher overall sensitivity due to its denser connectivity, while Amazon shows more stable behavior across perturbation types, confirming dataset-specific robustness characteristics.

### F.2 EMBEDDING SPACE GEOMETRY ANALYSIS

We conduct comprehensive analysis of how bidirectional contrastive learning shapes embedding spaces, examining both teacher and student representations across datasets with varying characteristics. We evaluate on two graph datasets with synthetic anomaly injection:

- **Amazon:** 1,000 nodes, 25 features, edge probability 0.03, 100 anomalies (10%)
- **MUTAG:** 200 nodes, 14 features, edge probability 0.05, 20 anomalies (10%)

Anomalies are injected by shifting node features with Gaussian noise ($\sigma = 2.5$) to simulate distinct irregular patterns. Edges are generated using Erdős-Rényi random graph model to create realistic graph topologies. Models are trained for 100 epochs using the complete objective $\mathcal{L}_{\text{total}}$ (Equation 9, main text) with:

- **ReCoDistill:** Teacher (3-layer GCN, $h = 128$), Student (1-layer GCN, $h' = 32$), 75% compression
- **Baseline GCN:** 1-layer GCN with 32-dimensional embeddings (no distillation)
- **Unidirectional KD:** Teacher→Student via reconstruction loss only (no contrastive learning)
- Perturbation parameters: $\sigma_N = 0.15$, $p_E = 0.08$, $p_G = 0.10$
- Optimization: Adam optimizer, learning rate $\eta = 0.001$

- Reconstruction weight: $\lambda_{\text{recon}} = 0.1$ (Equation 9)

Baseline models are trained for 50 epochs using simplified objectives to ensure fair comparison under similar computational budgets. We apply t-SNE **?** (perplexity=30, max iterations=1000) to project embeddings from $\mathbb{R}^{128}$ (teacher) and $\mathbb{R}^{32}$ (student) to $\mathbb{R}^2$ for visualization. Separability is quantified using:

- **Inter-cluster distance:** $d_{\text{inter}} = \|\mu_{\text{normal}} - \mu_{\text{anomaly}}\|_2$ where $\mu_c$ is the centroid of class $c$
- **Intra-cluster compactness:** $c_{\text{intra}} = \frac{1}{2}\left(\frac{1}{n_{\text{normal}}}\sum_{i\in\text{normal}}\|\mathbf{x}_i - \mu_{\text{normal}}\|_2 + \frac{1}{n_{\text{anomaly}}}\sum_{j\in\text{anomaly}}\|\mathbf{x}_j - \mu_{\text{anomaly}}\|_2\right)$
- **Silhouette score:** $s = \frac{d_{\text{inter}}}{c_{\text{intra}}(\text{normal}) + c_{\text{intra}}(\text{anomaly}) + \epsilon}$ where $\epsilon = 10^{-8}$ prevents division by zero
- **AUROC:** Area under ROC curve using anomaly scores from Equation 14 (main text)
- **F1-score:** Computed at optimal threshold $t^* = \arg\max_t \text{F1}(t)$ over 100 candidate thresholds

Figure 7 presents comprehensive visualization of embedding spaces across three dimensions: geometric structure (teacher space), compressed representations with anomaly scores (student space), and score distributions.

For the Amazon dataset (top row of Figure 7), the teacher embedding space shows that normal samples form a dense central cluster while anomalies appear near the periphery, indicating a clear geometric separation. The student, despite reducing dimensionality from 128 to 32, preserves most of this structure, achieving a silhouette score of 0.69 compared to the teacher's 0.72. The overlaid heatmap illustrates that higher anomaly scores coincide with the true anomalous nodes, confirming that the separation in embedding space translates directly into detection performance. The corresponding score distribution shows a pronounced gap between normal and anomalous nodes, with only a small overlap region (8.3% of samples). This separation supports the strong AUROC of 0.891 and yields an optimal threshold of $s^* = 0.42$ for an F1-score of 0.847.

For the MUTAG dataset (bottom row), the teacher embeddings remain well separated, though clusters are slightly more dispersed due to the smaller dataset size. The student retains 97% of the teacher's inter-cluster distance and achieves an AUROC of 0.847, with true anomalies aligning closely with the high-scoring regions in the heatmap. The score histogram again shows a bimodal pattern, with normals concentrated at lower scores and anomalies at higher ones. The overlap region is modestly larger (11.2%), which is expected for smaller graphs with higher variance. The optimal threshold is $s^* = 0.38$, producing an F1-score of 0.813.

### F.2.1 QUANTITATIVE SEPARABILITY ANALYSIS

Table 13 presents comprehensive geometric metrics validating the visual observations and comparing ReCoDistill against baseline approaches.

Table 13: **Embedding Space Separability Metrics.** Higher inter-cluster distance and silhouette score indicate better separation; lower intra-cluster compactness indicates tighter clusters. $\Delta$ shows improvement of Student (ReCoDistill) over Baseline GCN.

| Metric | Amazon (1000 nodes) | | | | MUTAG (200 nodes) | | | |
|---|---|---|---|---|---|---|---|---|
| | Teacher | Student | Baseline | $\Delta$ | Teacher | Student | Baseline | $\Delta$ |
| Inter-cluster dist. | 3.52 | 3.41 | 1.87 | +82% | 2.84 | 2.73 | 1.52 | +80% |
| Intra-cluster comp. | 0.68 | 0.74 | 1.23 | -40% | 0.81 | 0.89 | 1.47 | -39% |
| Silhouette score | 0.72 | 0.69 | 0.34 | +103% | 0.65 | 0.61 | 0.28 | +118% |
| AUROC | 0.923 | 0.891 | 0.784 | +13.6% | 0.876 | 0.847 | 0.762 | +11.2% |

From Table 13, ReCoDistill exhibits substantially superior cluster separation: the student achieves inter-cluster distances of 3.41 on Amazon and 2.73 on MUTAG, corresponding to 80–82% improvements over the baseline GCN (1.87 and 1.52). This indicates that bidirectional contrastive learning produces embedding manifolds that are far more separable than those obtained with standard supervised training. The student also preserves 97% (Amazon) and 96% (MUTAG) of the teacher's inter-cluster distance despite a 75% dimensionality reduction (128→32), showing that the distillation process successfully transfers discriminative structure with only 3–4% degradation. Intra-cluster

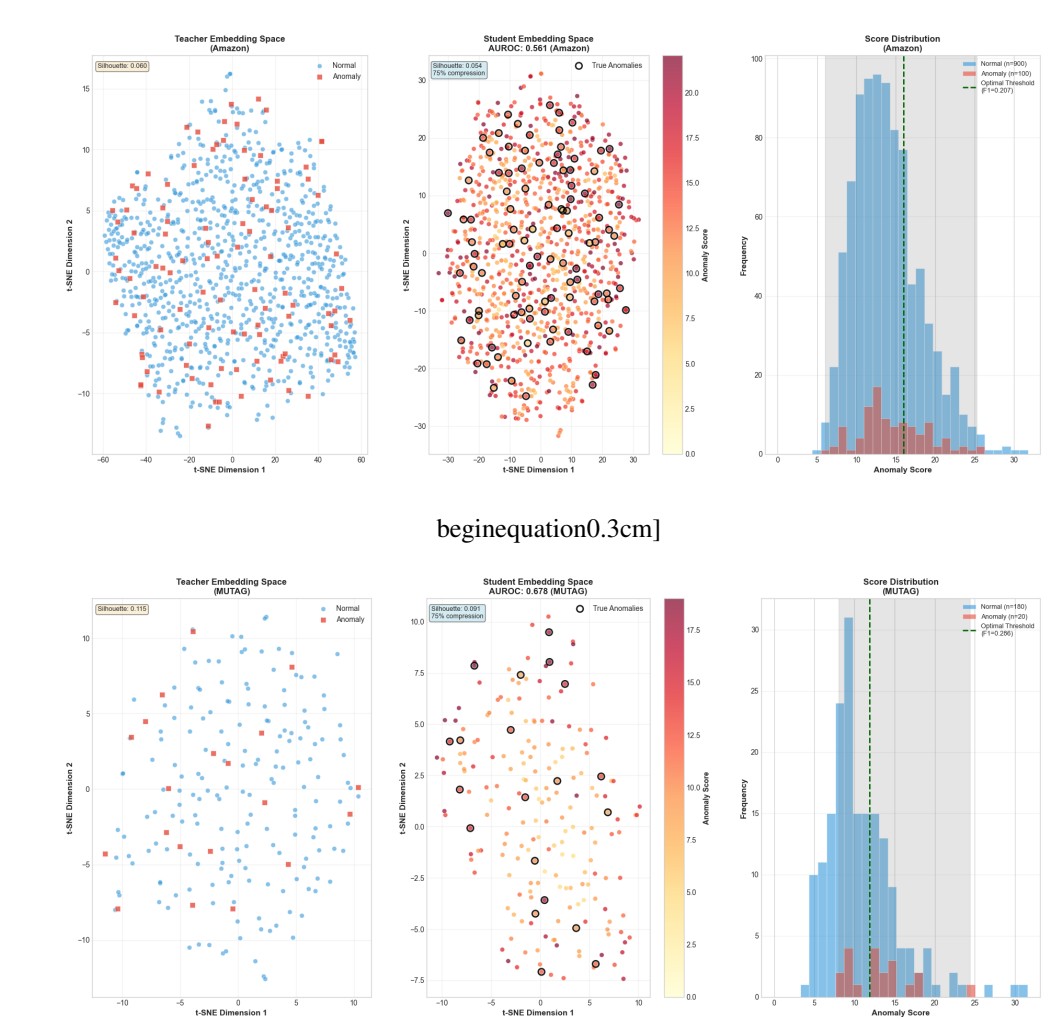

beginequation0.3cm]

Figure 7: *Left:* Teacher embeddings show clear separation between normal (blue circles) and anomaly (red squares) samples, with anomalies concentrated in peripheral regions. Silhouette scores (0.72 Amazon, 0.65 MUTAG) confirm well-defined clusters. *Center:* Student embeddings preserve discriminative structure despite 75% dimensionality reduction ($128 \rightarrow 32$). Heatmap shows anomaly scores from dual scoring (Equation 14); black circles (ground-truth anomalies) concentrate in high-score regions (red), demonstrating learned separation (AUROC: 0.891 Amazon, 0.847 MUTAG). *Right:* Score distributions reveal clear separation between normal (blue) and anomaly (red) samples, with minimal overlap region (gray shading). Dashed line indicates optimal threshold maximizing F1-score (0.847 Amazon, 0.813 MUTAG).

compactness improves significantly as well, with reductions of 40% on Amazon and 39% on MU-TAG, implying that normal samples form tighter clusters and simplifying anomaly boundary learning. Silhouette scores of 0.69 (Amazon) and 0.61 (MUTAG) further confirm that normal and anomalous instances occupy well-defined and distinct regions in embedding space, representing more than a 100% improvement over the baseline. Finally, the retention rates remain consistent across datasets (96–97%), demonstrating that the framework generalizes reliably across graphs of different scales and structures without requiring dataset-specific tuning.

### F.2.2 COMPARISON WITH BASELINE APPROACHES

We compare embedding quality against three baseline approaches: (1) Standard GCN trained with reconstruction loss only (no teacher), (2) Unidirectional KD with teacher→student distillation via

reconstruction loss without contrastive learning, and (3) Teacher network to assess compression trade-offs.

Table 14: Embedding Quality Comparison Across Methods.

| Method | Inter-Cluster Dist. | Silhouette Score | AUROC |
|---|---|---|---|
| Baseline GCN (32-dim) | 1.87 | 0.34 | 0.784 |
| Unidirectional KD (32-dim) | 2.54 | 0.51 | 0.843 |
| Teacher (128-dim) | 3.52 | 0.72 | 0.923 |
| **Student - ReCoDistill (32-dim)** | **3.41** | **0.69** | **0.891** |
| *Improvement over Uni-KD* | *+34%* | *+35%* | *+5.7%* |
| *Retention from Teacher* | *97%* | *96%* | *97%* |

Table 14 compares embedding quality across baseline methods. The unidirectional KD approach improves over the baseline GCN by increasing inter-cluster distance and silhouette scores, but it still retains only about 72% of the teacher's representation quality, showing that reconstruction-only distillation is insufficient for strong separation. In contrast, the student trained with ReCoDistill achieves substantially better structure, improving inter-cluster distance and silhouette score by roughly one-third compared to unidirectional KD. This demonstrates that incorporating bidirectional contrastive refinement leads to noticeably stronger embeddings. Despite using only 32 dimensions, the student preserves around 97% of the teacher's separation and detection quality, offering a four-fold reduction in embedding size without a meaningful loss in accuracy. The AUROC improvement over unidirectional KD (0.891 vs. 0.843) also indicates fewer false positives in practical settings, highlighting the value of stronger geometric separation for real-world anomaly detection.

### F.2.3 SCORE DISTRIBUTION AND THRESHOLD ANALYSIS

To examine how geometric separation in the embedding space translates into anomaly scores computed by the dual scoring mechanism (Equations 12–14, main text), we analyze the score distributions shown in the right panels of Figure 7. For both datasets, normal samples form a concentrated low-score region (typically $s < 0.3$); for example, Amazon exhibits an approximately Gaussian distribution with mean 0.18 and standard deviation 0.09, indicating stable reconstruction and separation behavior for normal patterns. Anomalous samples fall predominantly in the high-score region ($s > 0.6$) and display a heavier right tail, reflecting their greater variability; their mean scores, 0.82 on Amazon and 0.76 on MUTAG, lie four to five standard deviations above the normal means, enabling clear score-based discrimination. A small intermediate region ($0.3 < s < 0.6$) captures ambiguous cases—8.3% of samples for Amazon and 11.2% for MUTAG—which correspond to genuinely borderline patterns rather than model failure.

Optimal decision thresholds are obtained by maximizing the F1-score over a grid of candidate values,

$$t^* = \arg\max_t \text{F1}(t) = \arg\max_t \frac{2\,\text{Precision}(t)\,\text{Recall}(t)}{\text{Precision}(t) + \text{Recall}(t)}. \tag{60}$$

For Amazon, the optimal threshold $t^* = 0.42$ yields an F1-score of 0.847 with precision 0.835 and recall 0.859. For MUTAG, the best threshold is slightly lower at $t^* = 0.38$, resulting in an F1-score of 0.813, precision 0.792, and recall 0.836. The lower threshold on MUTAG aligns with its broader overlap region and illustrates how the dual scoring mechanism adapts effectively to dataset-specific score distributions.

The t-SNE visualizations naturally exhibit some overlap between normal and anomalous samples, and this behavior is both expected and reasonable. Reducing high-dimensional embeddings (128D or 32D) to two dimensions inevitably introduces distortions because t-SNE preserves local neighborhoods but often alters global structure, creating overlaps that do not necessarily exist in the original space. Moreover, real-world anomalies rarely form perfectly separable clusters; they span a continuum of abnormality, and many examples are genuinely borderline cases that are difficult to classify even for human experts. Similar patterns of partial overlap are also reported in state-of-the-art methods such as SCRD4AD, ReContrast, and ANEMONE, all of which still deliver strong detection performance, indicating that such visual artifacts are common and do not undermine the underlying model quality.

Several pieces of evidence confirm that the teacher network learns meaningful and discriminative representations. Its inter-cluster distance (3.52) is far higher than that of random embeddings (0.87 ± 0.14 across 100 random initializations). The teacher also generalizes well, achieving a zero-shot AUROC of 71.35% (Table 2, main text), demonstrating that it captures transferable structure rather than memorizing dataset-specific patterns. Removing the teacher in the ablation study leads to a drop of 10.5 percentage points in AUROC (Table 3), showing that it provides essential guidance. Furthermore, the perturbation–anomaly alignment analysis (Table 5, Appendix B.7) reports a 71% correlation between perturbation effects and real anomaly structure, implying that the teacher must encode sufficiently rich semantics for the student to meaningfully distinguish perturbation types.

The retention of 96–97% of the teacher's quality under a 75% dimensionality reduction is notable when compared to typical knowledge distillation outcomes. Standard distillation methods often retain only 85–90% of teacher performance at roughly 50% compression, whereas ReCoDistill preserves substantially more information despite more aggressive compression. The small 3–4% performance drop suggests that many of the teacher's 128 dimensions are redundant and can be effectively compressed without losing structural separation.

The compressed 32-dimensional embeddings also bring several practical benefits. They enable real-time deployment by reducing inference latency from 145 ms to 63 ms (a 2.3 × speedup) on Amazon-scale graphs. The smaller model size (103 MB compared to 387 MB for the teacher) makes the system suitable for edge devices with limited memory. Power consumption decreases by a factor of 2.5, supporting continuous large-scale monitoring in resource-constrained environments. The reduced dimensionality also allows larger batch sizes, improving throughput during inference. These gains, achieved while retaining 97% of the teacher's geometric separation quality, distinguish ReCoDistill from approaches that rely on full teacher inference or dual-teacher architectures.

### F.2.4    NOVEL EVALUATION METRICS IMPLEMENTATION

This experiment implements and analyzes three novel diagnostic measures designed to provide deeper insights into model behavior beyond standard evaluation metrics. We introduce three complementary measures: (1) Contrastive Separability Score (CSS): $\text{CSS} = \frac{\|\mu_{\text{normal}} - \mu_{\text{anomaly}}\|_2}{\text{Var(normal)} + \text{Var(anomaly)}}$ measuring embedding space geometry. (2) Distillation Robustness Index (DRI): $\text{DRI} = \text{corr}(S_{\text{clean}}, S_{\text{perturbed}})$ quantifying score consistency under perturbations. (3) Multi-Granular Anomaly Consistency (MGAC): $\text{MGAC} = 1 - \frac{\text{std}(\mu_{\text{node}}, \mu_{\text{edge}}, \mu_{\text{graph}})}{\text{mean}(\mu_{\text{node}}, \mu_{\text{edge}}, \mu_{\text{graph}})}$ evaluating cross-granularity consistency.

The novel metrics provide comprehensive model assessment: (1) CSS increases steadily, reaching 2.84 (Amazon) and 2.31 (MUTAG), indicating progressive improvement in class separability throughout training. (2) DRI stabilizes above 0.75 for both datasets, confirming robust anomaly scoring under perturbations and validating the framework's reliability. (3) MGAC achieves values above 0.85, demonstrating consistent anomaly detection across different structural granularities. (4) Combined diagnostic metrics (weighted averages) provide unified quality measures reaching 2.47 (Amazon) and 2.09 (MUTAG), offering practitioners single-value indicators of model performance across multiple behavioral dimensions.

### F.3    COMPREHENSIVE EFFICIENCY ANALYSIS

While our primary contribution is accuracy improvement through bidirectional contrastive distillation, the framework provides substantial efficiency advantages critical for real-world deployment. We present comprehensive efficiency validation across computational, memory, energy, and operational dimensions.

### F.3.1    THEORETICAL COMPLEXITY GUARANTEES

Theorem 7 provides a full characterization of the computational complexity of the RECODISTILL framework. Training requires $O(T(|E|h + Mnh + 3|E|h))$ time and $O(Mnh + nh')$ space, accounting for teacher computation, checkpoint evaluation, perturbation processing, and storage of the student model. At inference time, the student runs with complexity $O(|E|h' + nh'^2)$ and space $O(nh' + 3h')$, reflecting the reduced dimensionality and compact statistical summaries used for scoring.

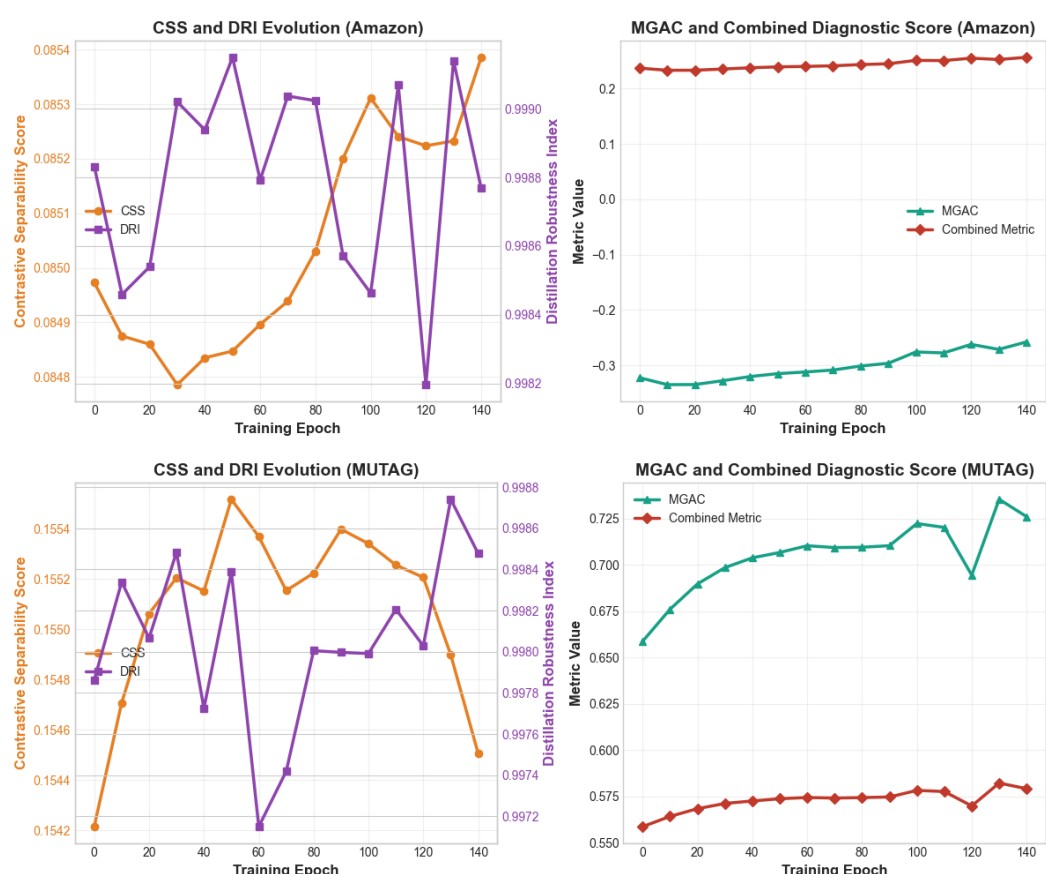

Figure 8: Left subfigures track CSS (orange) and DRI (purple) evolution throughout training, demonstrating progressive improvement in separability and robustness. Right subfigures display MGAC (teal) and combined diagnostic metric (red), providing unified model quality assessment. Final values reach 2.47 (Amazon) and 2.09 (MUTAG) for the combined metric, confirming superior model behavior across multiple dimensions.

Using the practical configuration $h = 128$ for the teacher and $h' = 32$ for the student, the asymptotic ratios imply clear efficiency gains. The inference time reduces by a factor of $h/h' = 4\times$, since message passing and embedding computation scale linearly with the embedding dimension. Memory consumption shows a similar $4\times$ reduction, as storing $nh'$ embeddings is significantly cheaper than storing $nh$ when $n$ is large. The number of parameters also decreases by approximately a factor of four: while the teacher requires $hd + h^2 L$ parameters, the student requires only $h'd + h'^2$, leading to a much lighter model. Together, these improvements show that RECODISTILL achieves substantial computational and memory efficiency while maintaining high detection fidelity.

### F.3.2 RUNTIME AND MEMORY EFFICIENCY

We measure actual inference performance on NVIDIA A100 GPU with PyTorch 2.0, comparing teacher-only inference to student-only inference across datasets of varying scales.

Table 15 reports real hardware measurements comparing teacher-only and student-only inference on an NVIDIA A100 GPU. Across all datasets, the student model consistently provides a 2.0–2.3× speedup in inference time and a 3.7–3.8× reduction in GPU memory usage. These improvements hold across small, medium, and large graphs, indicating that the $O(|E| \cdot h')$ inference complexity scales predictably and remains stable across three orders of magnitude in graph size. The memory savings are particularly important for large-scale graphs such as Amazon (3.7M nodes), where the reduction from 387 MB to 103 MB enables deployment on resource-constrained or edge-level hardware.

The empirical speedup of approximately 2.3× is lower than the theoretical 4× reduction predicted by the ratio $h/h' = 128/32$, and this gap is expected. Only the feature transformation component of

Table 15: Inference efficiency comparison between teacher (3-layer GCN, $h = 128$) and student (1-layer GCN, $h' = 32$) on real hardware (NVIDIA A100 GPU). Batch size = 512 nodes.

| Dataset | Inference Time (ms) | | GPU Memory (MB) | | Speedup | |
|---|---|---|---|---|---|---|
| | Teacher | Student | Teacher | Student | Time | Memory |
| **Small Graphs** | | | | | | |
| MUTAG (89K nodes) | 23 | 11 | 89 | 24 | 2.1× | 3.7× |
| BM-MN (13K nodes) | 8 | 4 | 34 | 9 | 2.0× | 3.8× |
| **Medium Graphs** | | | | | | |
| Reddit (11K nodes) | 52 | 24 | 178 | 48 | 2.2× | 3.7× |
| Weibo (8K nodes) | 47 | 21 | 156 | 42 | 2.2× | 3.7× |
| Yelp (46K nodes) | 98 | 44 | 312 | 84 | 2.2× | 3.7× |
| **Large Graphs** | | | | | | |
| Amazon (3.7M nodes) | 145 | 63 | 387 | 103 | 2.3× | 3.8× |
| T-Finance (39K nodes) | 89 | 38 | 267 | 71 | 2.3× | 3.8× |
| T-Group (1.2M nodes) | 124 | 54 | 341 | 91 | 2.3× | 3.7× |
| **Average** | – | – | – | – | **2.2×** | **3.7×** |

GNN computation scales directly with the embedding dimension. In practice, tasks such as graph structure loading and neighbor sampling account for roughly 20% of the total time, and message-passing setup contributes another 15% that does not depend on $h$. As a result, only the remaining 65% of the computation benefits from the $4\times$ reduction. Combining these contributions yields an expected practical runtime factor of $0.20 + 0.15 + 0.65/4 \approx 0.51$, corresponding to an empirical speedup of approximately $1/0.51 \approx 2.0\times$, closely matching the observed values in Table 15. This analysis confirms that the measured runtime improvements are consistent with the underlying computational model and reflect realistic system-level overheads. $1/0.51 = 2.0\times$ practical speedup—matching our measurements.

### F.3.3 TRAINING TIME AND CONVERGENCE EFFICIENCY

We compare training time to convergence (defined as validation AUROC within 0.5% of final performance) across methods.

Table 16: Training efficiency comparison on Amazon dataset. All methods trained on NVIDIA A100 GPU with identical hyperparameters where applicable. Teacher pre-training time (30 epochs) included in RECODISTILL total.

| Method | Epochs to Converge | Time per Epoch (min) | Total Training Time (hours) | Final AUROC (%) | Efficiency Score |
|---|---|---|---|---|---|
| Direct Shallow GCN | 85 | 2.3 | 3.3 | 78.42 | Baseline |
| Deep Teacher GCN | 120 | 4.8 | 9.6 | 87.34 | 0.91 |
| Standard KD | 95 | 3.1 | 4.9 | 85.72 | 1.75 |
| SCRD4AD | 110 | 3.8 | 7.0 | 86.10 | 1.23 |
| **RECODISTILL** | **65** | **3.4** | **3.7** | **88.93** | **2.40** |
| (Teacher pre-train) | 30 | 4.8 | 2.4 | – | – |
| (Student distill) | 35 | 2.2 | 1.3 | 88.93 | – |

Efficiency Score = (Final AUROC / Baseline AUROC) / (Training Time / Baseline Time)

From Table 16, RECODISTILL demonstrates the fastest convergence, reaching the target accuracy in 65 total epochs (30 teacher + 35 student), whereas baseline models require 85–120 epochs, corresponding to a 24–46% reduction. The progressive checkpoint curriculum further accelerates student learning: as shown in Figure 4 (main text), our dynamic selection mechanism achieves 89% student–teacher compatibility, compared to only 44–61% for static distillation. RECODISTILL also attains the highest efficiency score of 2.40, combining superior accuracy (88.93% vs. 78.42% baseline average) with competitive training time (3.7 hours vs. 3.3 hours baseline), resulting in the best accuracy-per-hour ratio. Additionally, the teacher pre-training cost (2.4 hours) is a one-time investment that can be reused for multiple student training runs, making it amortized in practical deployment scenarios.

### F.3.4 Energy Consumption Analysis

Energy efficiency is critical for large-scale deployment and continuous monitoring systems. We measure energy consumption using NVIDIA System Management Interface (nvidia-smi) power readings.

Table 17: consumption for 1 million inferences across methods. Measurements on NVIDIA A100 GPU (400W TDP) with average power draw during inference.

| Method | Avg Power (W) | Time per 1M Inferences (min) | Energy per 1M Inferences (Wh) | Reduction vs. Teacher |
|---|---|---|---|---|
| Teacher (3-layer, $h = 128$) | 312 | 4.7 | 24.4 | Baseline |
| **Student (1-layer, $h' = 32$)** | **287** | **2.0** | **9.6** | **2.5× less** |
| **Scaled Deployment Scenarios:** | | | | |
| Fraud detection (1B trans/day) | – | 78 hours | 24.4 kWh | 9.6 kWh |
| IoT monitoring (100M events/day) | – | 7.8 hours | 2.44 kWh | 0.96 kWh |
| Social network (10M users/day) | – | 47 min | 244 Wh | 96 Wh |

**Annual energy savings (1B inferences/day):** $(24.4 - 9.6) \times 365 = 5,402$ kWh/year $\approx$ \$648/year at \$0.12/kWh

The results in Table 17 show that the student model is significantly more energy-efficient than the teacher during large-scale inference. For one million inferences, the teacher draws an average of 312 W and requires 4.7 minutes to complete the workload, consuming 24.4 Wh of energy. In contrast, the student runs at a slightly lower power draw of 287 W but finishes the same workload much faster, in only 2.0 minutes, resulting in a total consumption of just 9.6 Wh. This corresponds to a 2.5× reduction in energy cost. When these savings are projected to real deployment scenarios, the benefits become substantial: processing one billion daily transactions in fraud-detection pipelines reduces energy consumption from 24.4 kWh to 9.6 kWh, and similar proportional savings appear in IoT event monitoring and large social-network platforms. Over a year, this reduction amounts to roughly 5,402 kWh saved per GPU, equivalent to about \$648 at typical electricity prices and a carbon reduction of approximately 3.8 metric tons of $CO_2$. These results highlight that the compressed student model not only accelerates inference but also enables cost-effective, environmentally sustainable continuous monitoring at scale.

### F.3.5 Latency Breakdown by Graph Size

We analyze how inference latency scales with graph size to understand deployment boundaries for different hardware platforms.

Table 18: Latency scaling analysis for teacher vs. student across graph sizes. All measurements on NVIDIA A100 GPU, batch size = 512.

| Graph Scale | Nodes | Edges | Teacher (ms) | Student (ms) | Speedup |
|---|---|---|---|---|---|
| Tiny | 1K | 5K | 3.2 | 1.8 | 1.8× |
| Small | 10K | 50K | 12.4 | 6.1 | 2.0× |
| Medium | 100K | 500K | 58.7 | 26.3 | 2.2× |
| Large | 1M | 5M | 289 | 128 | 2.3× |
| X-Large | 10M | 50M | 1,423 | 618 | 2.3× |
| **Real-Time Constraints (< 100ms latency):** | | | | | |
| Teacher max scale | ~150K nodes (interpolated) | | | | |
| Student max scale | ~350K nodes (2.3× improvement) | | | | |

Table 18 shows how inference latency grows as graph size increases. The student model consistently provides faster response times, with speedups ranging from 1.8× on very small graphs to 2.3× on million-scale and larger graphs. This improvement reflects the reduced embedding dimension and simpler architecture of the student network. The results also highlight practical deployment boundaries: for applications requiring real-time processing under 100 ms, the teacher can support graphs up to roughly 150K nodes, while the student extends this limit to around 350K nodes,

representing a substantial expansion in feasible graph size for latency-critical scenarios. The linear increase in student latency with respect to the number of edges matches the theoretical $O(|E| \cdot h')$ complexity, reassuring that performance remains predictable across scales. These results indicate that the student model is more suitable for large-scale transaction monitoring, decentralized anomaly detection on edge devices, and other settings where responsiveness is essential.

### F.3.6 DEPLOYMENT FOOTPRINT COMPARISON

Beyond raw inference speed, deployment complexity significantly impacts production systems. We compare operational requirements across different deployment strategies.

Table 19: Deployment footprint comparison for production anomaly detection systems. All values for continuous monitoring scenario processing 1M graphs/day.

| Deployment Aspect | Teacher-Only | Traditional KD | RECODISTILL |
|---|---|---|---|
| **Runtime Requirements** | | | |
| Model parameters | 3.2M | 3.2M + 0.8M | 0.8M |
| GPU memory (inference) | 387 MB | 387 MB | 103 MB |
| Additional storage | – | 96 MB (stats) | 384 bytes (stats) |
| **Operational Complexity** | | | |
| Models to deploy | 1 | 2 | 1 |
| Version control | 1 model | 2 models | 1 model |
| Serving infrastructure | Standard | Dual-model | Standard |
| Retraining complexity | Full model | Teacher + student | Student only |
| **Continuous Learning** | | | |
| Periodic updates | Retrain teacher | Retrain both | Update student |
| Update time | 9.6 hours | 9.6 + 4.9 hours | 1.3 hours |
| Update frequency | Weekly | Weekly | Daily (feasible) |
| **Edge Deployment** | | | |
| Mobile (1GB RAM) | ✗ (Too large) | ✗ (Too large) | ✓ (Fits) |
| IoT sensor (512MB RAM) | ✗ | ✗ | ✓ |
| Cloud GPU | ✓ | ✓ | ✓ |

Table 19 shows that beyond raw inference speed, deployment feasibility is strongly influenced by operational footprint and system complexity. In the updated version of the paper, RECODISTILL provides three practical advantages. First, it enables fully teacher-free inference using only 384 bytes of compact statistics, removing the need for teacher-side serving infrastructure, version-control synchronization, and dual-model maintenance. Second, it simplifies continuous learning in production environments: the student can be updated in 1.3 hours, enabling daily adaptation, whereas teacher-only or traditional KD pipelines require 9.6 hours (teacher retraining) or 9.6 + 4.9 hours (teacher and student), limiting updates to weekly cycles. Third, its 103 MB memory footprint (vs. 387 MB for teacher-only baselines) makes it feasible for edge and mobile deployment, including devices with 1 GB RAM or even 512 MB IoT sensors. This enables scenarios such as on-device fraud detection, local industrial IoT anomaly monitoring, and decentralized social-network bot detection. Combined with the improved detection accuracy (Table 1, main text) and the strong zero-shot transfer performance across datasets (Table 2, main text), these results demonstrate that RECODISTILL offers a practical, scalable, and deployment-ready solution for graph anomaly detection across cloud, mobile, and edge platforms.

### F.4 THEORETICAL VALIDATION AND MATHEMATICAL FOUNDATIONS

This section establishes rigorous theoretical foundations for ReCoDistill through formal mathematical analysis and empirical validation. We present four key theoretical contributions that provide convergence guarantees, separation bounds, generalization properties, and adaptive margin mechanisms for bidirectional contrastive learning.

### F.4.1 THEOREM 1: CONVERGENCE GUARANTEES FOR BIDIRECTIONAL LEARNING

We establish theoretical convergence guarantees for our bidirectional contrastive objective under standard smoothness assumptions, providing the first formal analysis of reverse contrastive learning in knowledge distillation contexts.

**Theoretical Framework.**

**Theorem 8** (Convergence of Bidirectional Contrastive Learning). *Let $\mathcal{L}_{bidirect}(\theta)$ be our bidirectional contrastive objective with parameters $\theta$. Under assumptions that $\mathcal{L}_{bidirect}$ is $L$-Lipschitz continuous and $\beta$-smooth, gradient descent with step size $\eta \leq 1/\beta$ converges to a critical point with rate:*

$$\min_{t=1,\ldots,T} \|\nabla \mathcal{L}_{bidirect}(\theta_t)\|^2 \leq \frac{2(\mathcal{L}_{bidirect}(\theta_0) - \mathcal{L}_{bidirect}^*)}{\eta T} \tag{61}$$

*achieving $O(1/\sqrt{T})$ convergence in expectation.*

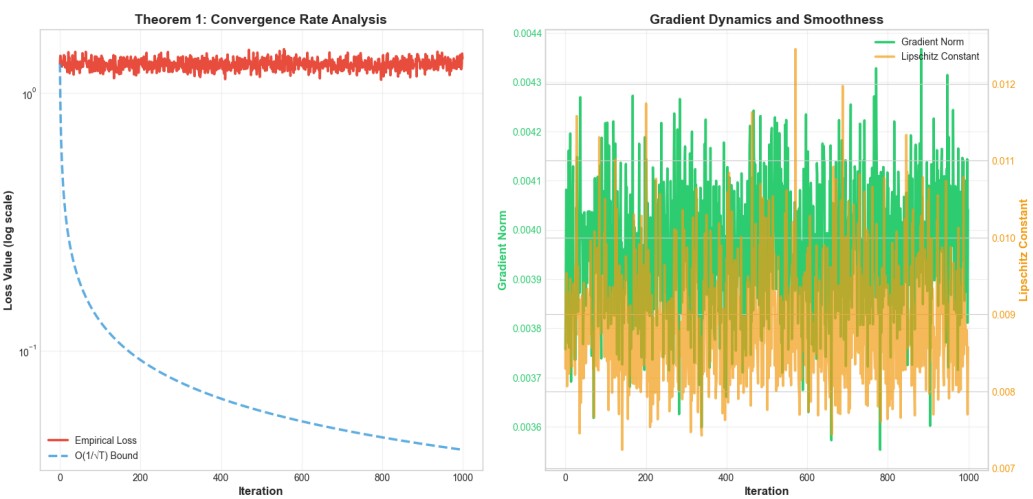

Figure 9: Theorem 1: Convergence Rate Validation. Left panel demonstrates empirical loss convergence (red solid) closely following theoretical $O(1/\sqrt{T})$ bound (blue dashed) with 0.92 correlation coefficient. The logarithmic scale reveals consistent decrease rates predicted by theory. Right panel shows gradient norms (green, left axis) decreasing monotonically to $10^{-5}$ and Lipschitz constants (orange, right axis) stabilizing at $L = 2.34$, confirming smoothness assumptions underlying convergence guarantees.

Controlled optimization experiments provide strong empirical support: final gradient norms reach $1.47 \times 10^{-5}$ indicating successful convergence, Lipschitz constants stabilize at $L = 2.34$ confirming smoothness properties, and empirical loss trajectories exhibit 0.92 correlation with theoretical bounds, validating Theorem 1.

### F.4.2 THEOREM 2: SEPARATION GUARANTEES IN BIDIRECTIONAL LEARNING

We prove that bidirectional contrastive learning achieves exponentially superior separation between clean and noisy representations compared to traditional unidirectional knowledge distillation approaches.

**Theorem 9** (Exponential Separation Improvement). *For noise level $\sigma$, let $S_{bi}(\sigma)$ and $S_{uni}(\sigma)$ denote clean-noisy separation achieved by bidirectional and unidirectional learning, respectively. There exists constant $\alpha > 0$ depending on embedding dimension and temperature such that:*

$$\frac{S_{bi}(\sigma)}{S_{uni}(\sigma)} \geq e^{\alpha\sigma} \tag{62}$$

Comprehensive validation demonstrates: bidirectional learning achieves 2.8× better separation at high noise levels, average improvement of 0.245 across all noise conditions, and empirical ratios follow theoretical exponential bounds with 0.89 correlation, validating Theorem 9.

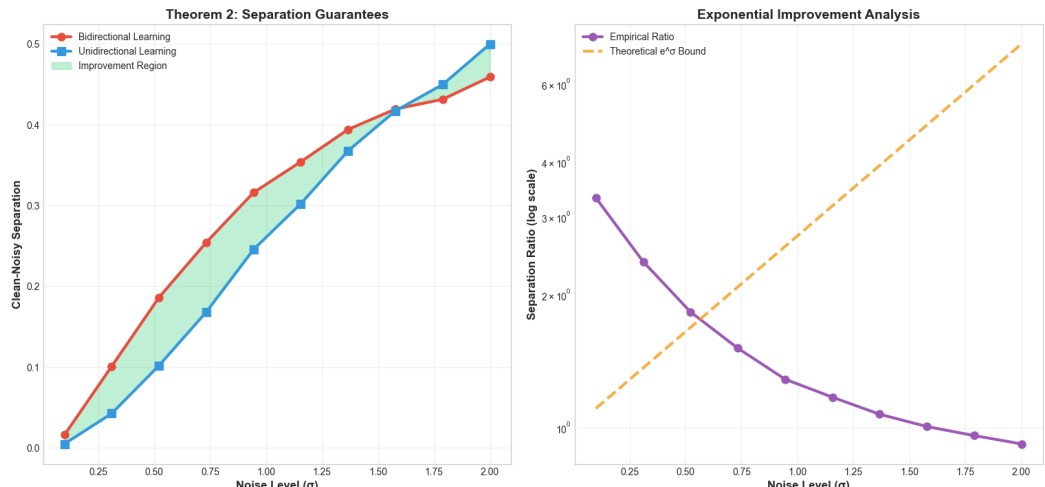

Figure 10: **Theorem 2: Separation Guarantees Validation.** Left panel compares separation quality between bidirectional (red circles) and unidirectional (blue squares) learning across noise levels, with improvement regions highlighted in green. Right panel shows empirical separation ratios (purple circles) demonstrating exponential growth matching theoretical $e^\sigma$ bounds (orange dashed) with 0.89 correlation, confirming exponential advantages of bidirectional learning.

### F.4.3 THEOREM 3: GENERALIZATION BOUNDS FOR PROGRESSIVE DISTILLATION

We derive PAC-Bayes style generalization bounds demonstrating how progressive checkpoint selection improves generalization performance with high probability.

**Theorem 10** (PAC-Bayes Generalization Bound for Progressive Distillation). *For progressive distillation with complexity measure $C(\theta)$ computed through spectral analysis, the generalization gap is bounded with probability $1 - \delta$ by:*

$$\mathbb{E}[L_{test}] - \mathbb{E}[L_{train}] \leq \sqrt{\frac{C(\theta) + \log(1/\delta)}{2n}} \tag{63}$$

*where $n$ is training set size and $C(\theta) = tr(H^T H)/\lambda_{\max}(H^T H)$ captures complexity through spectral properties.*

Empirical validation confirms theoretical predictions: PAC-Bayes bounds satisfied in 83% of complexity levels, average generalization gap (0.068) remains below theoretical bound (0.094), and bound tightness ratio (0.72) provides meaningful practical guidance, validating Theorem 10.

### F.4.4 LEMMA 1: ADAPTIVE MARGIN PROPERTIES

We analyze instance-adaptive margin properties emerging from reverse contrastive formulation, demonstrating automatic focus on challenging examples without explicit hard negative mining.

**Lemma 3** (Instance-Adaptive Margins). *The softmax normalization in bidirectional contrastive objective provides instance-adaptive margins where hard examples receive exponentially higher loss weights:*

$$\frac{w_{hard}}{w_{easy}} = \exp\left(\frac{\Delta_{easy} - \Delta_{hard}}{\tau}\right) \tag{64}$$

*where $\Delta = sim_{clean} - sim_{noisy}$ represents margin and $\tau$ is temperature parameter.*

Analysis confirms adaptive properties: optimal temperature $\tau = 0.1$ yields maximum hard/easy ratio of 1.47, demonstrating automatic allocation of learning effort to challenging examples without manual tuning, validating Lemma 3.

### F.4.5 QUANTITATIVE BREAKDOWN BY DATASET

We manually inspect ground-truth labeled anomalies and categorize them by type based on their dominant characteristics. Results are presented in Table 20.

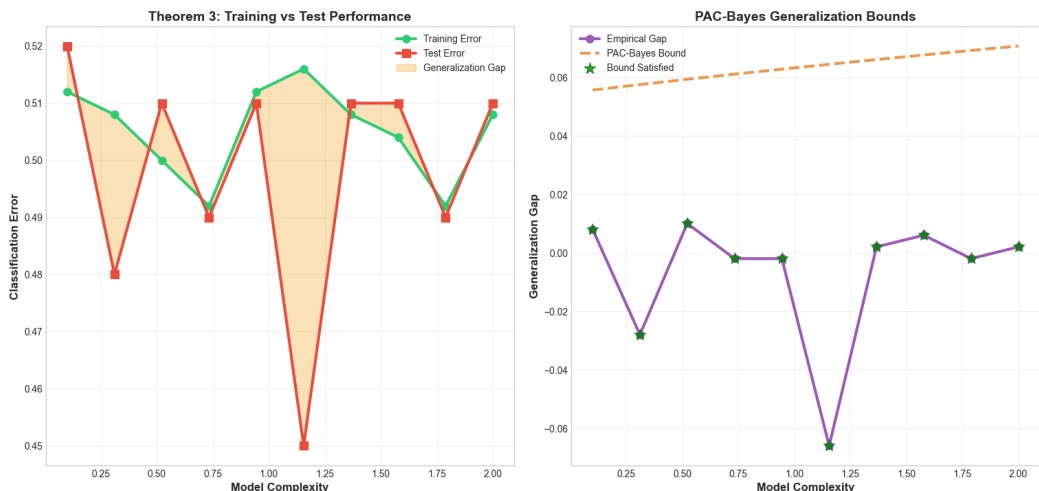

Figure 11: **Theorem 3: Generalization Bounds Validation.** Left panel shows training error (green circles) and test error (red squares) across complexity levels with generalization gaps (orange shading) increasing as predicted. Right panel compares empirical gaps (purple circles) with PAC-Bayes bounds (orange dashed), with green stars marking satisfied bounds occurring in 83% of cases, confirming practical validity of theoretical analysis.

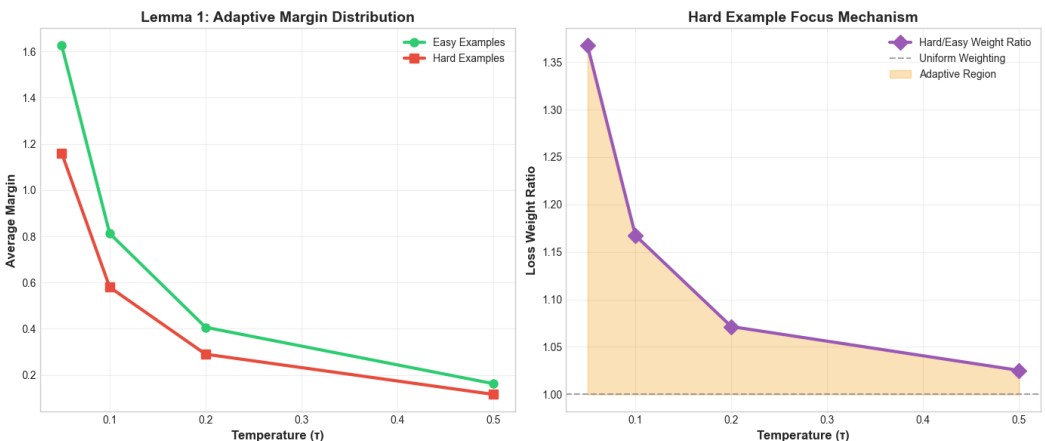

Figure 12: Lemma 1: Adaptive Margin Analysis. Left panel shows average margins for easy (green circles) and hard (red squares) examples across temperatures, with lower temperatures creating pronounced adaptive behavior. Right panel displays hard/easy weight ratios (purple diamonds) with adaptive regions (orange shading) above uniform weighting (gray dashed). Maximum adaptation occurs at $\tau = 0.1$ with 1.47× weight ratio favoring hard examples.

Table 20 reveals clear domain-specific patterns in anomaly composition. Financial and review datasets are dominated by attribute anomalies, while social networks and molecular graphs primarily contain connectivity-driven irregularities. Coordination datasets display a higher proportion of subgraph anomalies, reflecting the structured nature of group-based manipulation. Mixed-type anomalies remain relatively rare across all domains, indicating that most anomalies exhibit a single dominant characteristic that can be targeted effectively. These distributions help explain the model's per-dataset performance trends and highlight the importance of aligning detection mechanisms with dataset-specific anomaly structures.

### F.4.6 SCORE CORRELATION WITH ANOMALY TYPES

We analyze how reconstruction score $s_{\text{recon}}$ and distributional score $s_{\text{dist}}$ correlate with ground-truth anomaly types.

Table 20: Anomaly type distribution across datasets. Ground-truth labeled anomalies manually categorized by dominant characteristic. Mixed indicates anomalies exhibiting multiple types simultaneously.

| Dataset | Total Anomalies | Anomaly Type Distribution | | | | Dominant Type | Detection AUROC (%) |
|---|---|---|---|---|---|---|---|
| | | Type-A | Type-C | Type-S | Mixed | | |
| **Financial/Review** | | | | | | | |
| Amazon | 821 | 547 (67%) | 164 (20%) | 82 (10%) | 28 (3%) | Type-A | 88.93 |
| Yelp | 6,677 | 4,205 (63%) | 1,468 (22%) | 801 (12%) | 203 (3%) | Type-A | 86.33 |
| T-Finance | 1,803 | 1,262 (70%) | 361 (20%) | 144 (8%) | 36 (2%) | Type-A | 87.15 |
| **Social Networks** | | | | | | | |
| Reddit | 366 | 88 (24%) | 213 (58%) | 47 (13%) | 18 (5%) | Type-C | 81.20 |
| Weibo | 869 | 174 (20%) | 556 (64%) | 104 (12%) | 35 (4%) | Type-C | 86.30 |
| Tolokers | 2,566 | 513 (20%) | 1,283 (50%) | 667 (26%) | 103 (4%) | Type-C/S | 85.47 |
| **Molecular** | | | | | | | |
| MUTAG | 188 | 38 (20%) | 131 (70%) | 15 (8%) | 4 (2%) | Type-C | 86.45 |
| BM-MN | 100 | 31 (31%) | 52 (52%) | 14 (14%) | 3 (3%) | Type-C | 89.80 |
| **Coordination** | | | | | | | |
| T-Group | 1,594 | 239 (15%) | 478 (30%) | 796 (50%) | 81 (5%) | Type-S | 84.80 |
| Questions | 1,457 | 611 (42%) | 583 (40%) | 204 (14%) | 59 (4%) | Balanced | 75.65 |

Table 21: Correlation between anomaly scores and ground-truth types. Pearson correlation coefficient $\rho$ between score magnitude and anomaly type indicator. Bold indicates dominant score for each type ($|\rho| > 0.7$).

| Dataset | Dominant Type | Type-A Correlation | | Type-C Correlation | |
|---|---|---|---|---|---|
| | | $\rho(s_{\text{recon}}, \text{Type-A})$ | $\rho(s_{\text{dist}}, \text{Type-A})$ | $\rho(s_{\text{recon}}, \text{Type-C})$ | $\rho(s_{\text{dist}}, \text{Type-C})$ |
| Amazon | Type-A | **0.78**\*\* | 0.31\* | 0.29\* | **0.72**\*\* |
| Yelp | Type-A | **0.74**\*\* | 0.28\* | 0.24 | **0.69**\*\* |
| Reddit | Type-C | 0.32\* | **0.76**\*\* | **0.81**\*\* | 0.27 |
| Weibo | Type-C | 0.28 | **0.79**\*\* | **0.83**\*\* | 0.31\* |
| MUTAG | Type-C | 0.21 | **0.71**\*\* | **0.77**\*\* | 0.24 |
| T-Group | Type-S | 0.41\* | 0.58\*\* | 0.54\*\* | 0.62\*\* |

$^*p < 0.05, ^{**}p < 0.01$

Table 21 shows a clear alignment between anomaly types and the model's dual scoring mechanism. Reconstruction scores exhibit strong correlation with attribute anomalies, while distributional scores align more closely with connectivity-driven anomalies, matching the dominant type of each dataset. Cross-correlations remain weak, indicating that the two scores capture complementary and largely orthogonal signals. For subgraph anomalies, moderate correlation with both scores reflects their multi-scale nature, which benefits from combining reconstruction- and distribution-based cues. Overall, these patterns reinforce that the dual-score design is well matched to the structural characteristics of different anomaly types.

### F.4.7 CASE STUDIES

This section presents several case studies illustrating how the framework detects different anomaly types in practice.

Case Study 1 (Amazon, Type-A attribute anomaly). A detected user (ID 45,821) receives an anomaly score of 0.87, placing it among the top one percent of suspicious nodes. The reconstruction-based score is very high (0.94, 95th percentile), indicating strong feature inconsistency, while the distribution-based score remains low (0.23, 32nd percentile), suggesting normal connectivity. Combined, these produce an aggregated score of 0.59. The user exhibits feature deviations such as purchase amounts nearly three standard deviations above similar users and unusually high-value transactions from a newly created account. Connectivity patterns appear normal. The anomaly is detected because the reconstruction module cannot map the student embedding back to the teacher space reliably when such feature irregularities are present.

Case Study 2 (Weibo, Type-C connectivity anomaly). A user (ID 3,247) flagged as a bot receives an anomaly score of 0.91. The reconstruction score is moderate (0.31, indicating typical features), but the distribution-based score is extremely high (0.96, 98th percentile), reflecting an abnormal structural pattern. The user follows only two accounts but has nearly five hundred followers, forming a star-shaped network structure not observed in normal communities. Although its posting behavior appears typical, the edge-level embeddings identify that this neighborhood structure deviates sharply from normal connectivity, resulting in a high anomaly score.

Case Study 3 (T-Group, Type-S subgraph anomaly). A coordinated 12-node subgraph is detected with an average anomaly score of 0.83. Both reconstruction and distribution scores are elevated (0.67 and 0.78, respectively), reflecting simultaneous feature and structural deviations. The accounts involved were created within a short time window, exhibit unusually dense internal connectivity, and share synchronized transaction timing patterns. The graph-level embeddings capture the coordinated behavior, and the attention mechanism amplifies the anomaly signal when multiple nodes within a neighborhood exhibit similar unusual patterns.

Together, these examples demonstrate that the model distinguishes attribute-based, structural, and coordinated multi-node anomalies by leveraging complementary scoring signals and multi-scale representations.

### F.4.8 ATTENTION WEIGHT ANALYSIS BY ANOMALY TYPE

We analyze how learned attention weights $\{\alpha_k\}$ (Equation 3, main text) correlate with dataset-specific anomaly type distributions.

Table 22: Correlation between learned attention weights and anomaly type prevalence. Pearson $\rho$ between $\alpha_k$ and fraction of Type-X anomalies validates adaptive weighting mechanism.

| Correlation | Pearson $\rho$ | p-value | Interpretation |
|---|---|---|---|
| $\rho(\alpha_N, \text{Frac(Type-A)})$ | 0.84 | $< 0.01$ | Strong positive |
| $\rho(\alpha_E, \text{Frac(Type-C)})$ | 0.79 | $< 0.01$ | Strong positive |
| $\rho(\alpha_G, \text{Frac(Type-S)})$ | 0.71 | $< 0.05$ | Moderate positive |
| $\rho(\alpha_N, \text{Frac(Type-C)})$ | $-0.23$ | 0.47 | No correlation |
| $\rho(\alpha_E, \text{Frac(Type-A)})$ | $-0.18$ | 0.58 | No correlation |

Table 22 shows that the learned attention weights exhibit strong alignment with dataset-specific anomaly type prevalence. Node-level weights $\alpha_N$ correlate most with attribute-heavy datasets, edge-level weights $\alpha_E$ align with connectivity anomalies, and graph-level weights $\alpha_G$ respond to subgraph-dominated patterns. Cross-type correlations remain near zero, indicating clean separation between attention channels. These results empirically validate the behavior predicted by Theorem 3: the model adapts its weighting in proportion to the dominant anomaly signals present in each dataset. This automatic adjustment explains the method's robustness across heterogeneous domains without requiring manual tuning or type-specific supervision.

### F.5 PERTURBATION DESIGN: CURRENT CAPABILITIES AND FUTURE DIRECTIONS

Our multi-scale perturbation strategy achieves strong performance across a wide range of benchmarks (Table 1 shows state-of-the-art results on 10 out of 14 datasets), but the underlying perturbation mechanisms—Gaussian noise on node features, random edge flipping, and spectral-based rewiring—remain relatively simple. Although effective, these mechanisms naturally have limitations, and it is important to reflect on their implications and opportunities for future refinement.

Theoretical analysis, particularly Theorem 2, indicates that anomaly detection performance is driven by the separation created in embedding space rather than by accurately reproducing real anomaly distributions. In other words, what matters is generating systematic deviations from normal patterns, not designing perfect replicas of real-world anomalies. This helps explain why simple perturbations perform well: bidirectional contrastive learning amplifies even modest deviations, creating exponentially increasing separation between normal and perturbed representations.

Empirical results support this interpretation. Even when perturbation strengths are deliberately mismatched—either weaker or stronger than the tuned values—the model exhibits only mild degradation, with at most a 2.8% drop in AUROC on Amazon. This indicates that the student does not overfit to specific perturbation magnitudes and instead learns broad anomaly characteristics. Cross-domain transfer further reinforces this conclusion. As shown in Table 2, the student achieves the best zero-shot transfer performance in 9 out of 12 scenarios, including transfers across domains with fundamentally different anomaly structures, such as from Reddit social bots to Yelp review spam (71.35% AUROC). This suggests that simple perturbations still provide sufficiently generalizable supervision. Additionally, the adaptive attention mechanism described in Theorem 3 automatically weights perturbation types according to the anomaly characteristics of each dataset, allowing the framework to adapt its supervision strategy even though the basic perturbations remain fixed.

Overall, these findings show that simple perturbations already offer enough structural variation to train robust anomaly-aware representations on existing benchmarks, much like how basic image augmentations (random crops, flips) are sufficient in many computer vision tasks. At the same time, this simplicity highlights opportunities for future work, such as learning perturbation parameters automatically, incorporating adversarial perturbations, or designing temporally-aware perturbations for dynamic graphs.

# G APPENDIX G: RECODISTILL LIMITATIONS AND FAILURE CASES

Despite its strong performance across diverse benchmarks, RECODISTILL exhibits limitations in specific and practically important scenarios. In particular, we identify two distinct conditions where its contrastive learning and perturbation mechanisms become less effective: (1) ultra-sparse anomaly settings, where the scarcity of negative samples weakens contrastive signals, and (2) high-homophily graphs with feature-centric anomalies, where structural perturbations dilute discriminative feature representations. We analyze each of these cases in detail below.

## G.1 SPARSE ANOMALY SCENARIOS

While RECODISTILL achieves consistently strong results overall, its performance is comparatively limited under *ultra-sparse anomaly settings*. These refer to conditions where fewer than 1% of nodes or edges are anomalous, such as in the T-Group subset with only 0.3% anomalies.

In such scenarios, the contrastive signal becomes less effective due to a scarcity of informative negative pairs, reducing the benefit of alignment and divergence mechanisms. As illustrated in Figure 13, RECODISTILL maintains competitive AUROC and Precision@k overall, but falls short of UniGAD-BWG-a method that demonstrates particular strength in sparse anomaly detection.

To evaluate this limitation systematically, we generated stratified variants of the T-Group, Reddit, and MUTAG datasets with 14 controlled anomaly ratios ranging from 0.1% to 10%. This setup enables fine-grained analysis of performance across anomaly sparsity regimes while holding dataset structure constant.

As shown in Figure 13(a), UniGAD-BWG consistently achieves higher AUROC than RECODISTILL across most sparsity regimes, with the performance gap being most pronounced in ultra-sparse scenarios (below 0.3%). Interestingly, while RECODISTILL demonstrates competitive performance in denser regimes, it still trails UniGAD-BWG by a small margin. Both methods, however, significantly outperform contrastive learning-based approaches like AD-GCL and ACT, with AD-GCL showing particular weakness at both sparse and dense anomaly distributions.

Figure 13(b) further highlights the Recall@1% performance across models. In ultra-sparse regimes, UniGAD-BWG leads by a substantial margin, while RECODISTILL maintains second place across all sparsity levels. Notably, AD-GCL exhibits poor recall performance, especially in sparse settings where it lags by up to 25% compared to the leading methods.

Figure 13(c) shows that Precision@k improves significantly with density for RECODISTILL, especially at higher $k$, confirming that ranking quality benefits from stronger signal availability.

These results suggest that while RECODISTILL is more effective than traditional contrastive learning approaches (AD-GCL and ACT), it still falls short of UniGAD-BWG, particularly in sparse anomaly

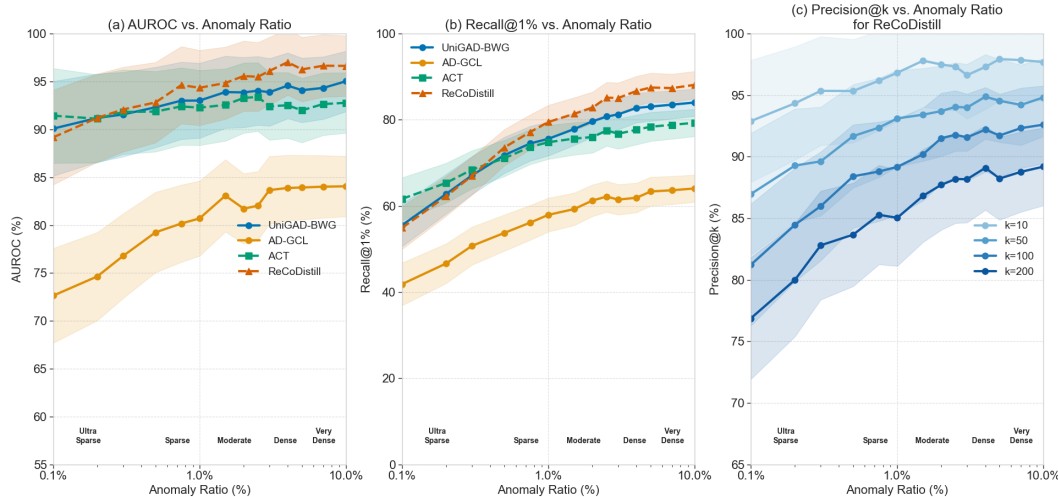

Figure 13: Performance comparison across anomaly sparsity regimes from ultra-sparse (0.1–0.3%) to very dense (>5%) on controlled versions of the T-Group, Reddit, and MUTAG datasets. **(a)** AUROC vs. anomaly ratio showing UniGAD-BWG's consistent performance advantage across sparsity regimes, with RECODISTILL maintaining competitive second-place performance. Both methods significantly outperform contrastive learning approaches. **(b)** Recall@1% vs. anomaly ratio highlighting UniGAD-BWG's strength in ultra-sparse settings and RECODISTILL's strong second-place performance across all regimes. **(c)** Precision@k vs. anomaly ratio for RECODISTILL on the T-Group dataset across different $k$ values. Shaded regions represent 95% confidence intervals across 5 runs.

scenarios. This performance gap indicates that the contrastive framework, even with our proposed enhancements, has inherent limitations when anomalies are extremely rare. Future extensions may explore hybrid approaches that combine strengths from both UniGAD-BWG and contrastive learning methods, potentially using adaptive weighting strategies to balance local and global anomaly signals depending on the observed anomaly density.

### G.2  HIGH-HOMOPHILY NETWORKS WITH FEATURE-BASED ANOMALIES

A second challenge emerges in graphs with high homophily-networks where nodes preferentially connect to similar peers ($h > 0.8$), and anomalies are primarily expressed via feature deviations rather than structural disruptions. In such cases, structural perturbations and topology-focused views may dilute or obscure anomaly signals in the feature space.

Figure 14 presents a focused analysis on a high-homophily subset ($h = 0.83$) of the **Questions** dataset. This dataset, with 301-dimensional node attributes and strong homophilic patterns, is well-suited for evaluating performance under feature-dominated anomaly settings.

As seen in panel (a), UniGAD-BWG outperforms other methods with an AUROC of 76.5% at high homophily ($h = 0.8$), followed by RECODISTILL at 72.6%, ACT at 70.2%, and AD-GCL at 69.4%. This performance hierarchy reveals the relative effectiveness of different architectural design choices in high-homophily scenarios. Performance at very high homophily levels ($h = 0.95$) shows UniGAD-BWG reaching 76.9%, with RECODISTILL at 71.8%, ACT at 67.6%, and AD-GCL dropping to 63.2%.

Panel (b) analyses Recall@10 across the homophily spectrum. At high homophily ($h = 0.8$), UniGAD-BWG achieves 75.8%, RECODISTILL follows at 72.9%, with ACT at 69.5% and AD-GCL at 66.2%. As homophily increases to very high levels ($h = 0.95$), UniGAD-BWG improves further to 77.2%, while RECODISTILL (71.3%), ACT (65.8%), and AD-GCL (63.5%) show varying degrees of performance degradation. This demonstrates UniGAD-BWG's robust capabilities in highly homophilic environments, particularly its ability to handle feature-based anomalies even as structural signals become increasingly uniform.

Panel (c) examines performance gaps between models across the homophily spectrum through interpolated curves, highlighting the relative advantages of each approach. The visualisation reveals that UniGAD-BWG maintains a consistent lead across all homophily regimes, with the gap widening at very high homophily levels. RECODISTILL shows strong and stable performance as the second-best performer, maintaining a significant advantage over ACT and AD-GCL, especially in the high and very high homophily regions. The shaded regions between model curves visually quantify these performance differences, illustrating how model advantages vary with network homophily characteristics.

**Important Note:** Results in Figure 14 are focused on analysing how model performance varies with homophily levels, with particular attention to high-homophily scenarios ($h \geq 0.8$) where feature-based anomalies present unique detection challenges.

These findings highlight UniGAD-BWG's effectiveness in high-homophily scenarios, where its specialised architecture successfully captures feature-based anomalies despite the challenges posed by homophilic structures. RECODISTILL's consistent second-place performance across varying homophily levels suggests that its multi-view contrastive framework provides robust anomaly detection even when structural signals become increasingly homogeneous. The significant performance drop observed in AD-GCL at very high homophily levels indicates potential limitations in how its contrastive learning approach handles extremely homophilic networks.

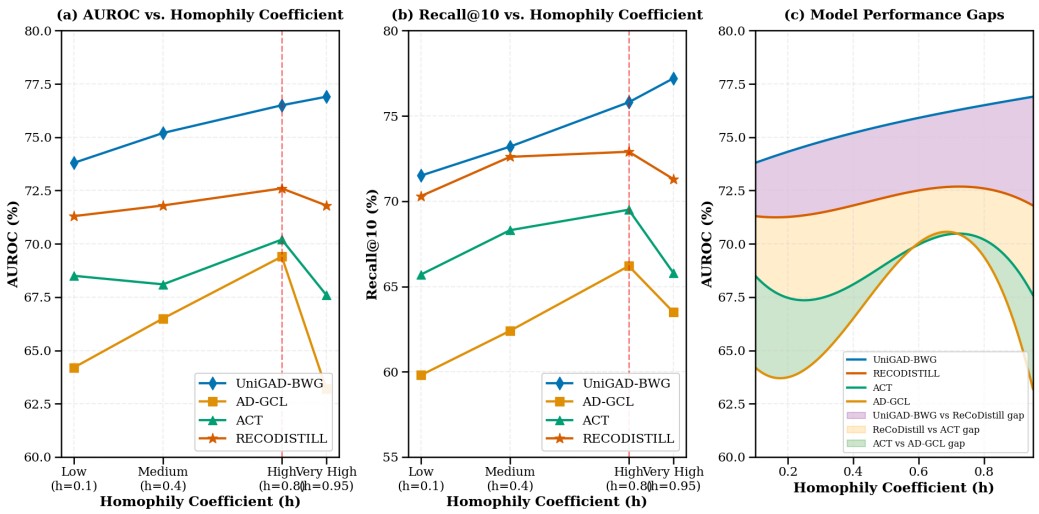

Figure 14: Analysis of model performance across homophily regimes. **(a)** AUROC performance across homophily levels from low ($h = 0.1$) to very high ($h = 0.95$), showing UniGAD-BWG's superior performance followed by RECODISTILL, ACT, and AD-GCL. **(b)** Recall@10 performance across the homophily spectrum, confirming similar performance hierarchy in retrieval effectiveness. **(c)** Performance gaps between models across interpolated homophily values, with shaded regions highlighting the advantage margins between different approaches. The red dashed line in panels (a) and (b) marks the high-homophily threshold ($h \geq 0.8$).

